# Exploiting Open-World Data for Adaptive Continual Learning

## Abstract

Continual learning (CL), which involves learning from sequential tasks without forgetting, is mainly explored in supervised learning settings where all data are labeled. However, high-quality labeled data may not be readily available at a large scale due to high labeling costs, making the application of existing CL methods in real-world scenarios challenging. In this paper, we study a more practical facet of CL: open-world continual learning, where the training data comes from the open-world dataset and is partially labeled and non-i.i.d. Building on the insight that task shifts in CL can be viewed as distribution transitions from known classes to novel classes, we propose OpenACL, a method that explicitly leverages novel classes in unlabeled data to enhance continual learning. Specifically, OpenACL considers novel classes within open-world data as potential classes for upcoming tasks and mines the underlying pattern from them to empower the model's adaptability to upcoming tasks. Furthermore, learning from extensive unlabeled data also helps to tackle the issue of catastrophic forgetting. Extensive experiments validate the effectiveness of OpenACL and show the benefit of learning from open-world data.[1]

## 1 Introduction

Continual learning (CL), unlike conventional supervised learning which learns from independent and identically distributed (i.i.d.) data, allows machines to continuously learn a model from a stream of data with incremental class labels. One of the main challenges in CL is to tackle the issue of the *catastrophic forgetting*, i.e., prevent forgetting the old knowledge as the model is learned on new tasks (De Lange et al., 2021). Although many approaches (e.g., data replay (Rebuffi et al., 2017; Lopez-Paz & Ranzato, 2017), weight regularization (Kirkpatrick et al., 2017; Li & Hoiem, 2017)) have been proposed to tackle catastrophic forgetting in CL, they rely on an assumption that a *complete* set of *labeled* data is available for training and focus on a supervised learning setting. Unfortunately, this assumption may not hold easily in real applications when obtaining high-quality sample-label pairs is difficult, possibly due to high time/labor costs, data privacy concerns, lack of data sources, etc. This is particularly the case for CL where the number of classes increases during the learning process.

To effectively learn CL models from limited labeled data, recent studies (Smith et al., 2021; Wang et al., 2021; Lee et al., 2019) suggest leveraging the semi-supervised learning (SSL) technique for CL to learn from both labeled and unlabeled data. The idea of SSL is to improve model performance by using limited labeled data and a larger amount of unlabeled data. In real applications, obtaining a steady stream of labeled data can be very expensive and time-consuming for CL, especially in new or rapidly evolving domains. However, obtaining large amounts of unlabeled data is relatively easier. SSL has proven effective and is applied to many tasks including CL. Specifically, Wang et al. (2021) considers a SSL setting where labeled and unlabeled data are assumed to be i.i.d. so that the unlabeled data can be leveraged to help improve the model performance. However, the i.i.d. assumption is commonly violated as the unlabeled data are usually acquired from different sources and distributional shifts exist between unlabeled and labeled data. In a worse case, the unlabeled data may be of low quality and contain large proportions of unknown data that do not

---

[1]Code available at https://anonymous.4open.science/r/openacl-5C3B/

belong to the classes of CL tasks. For example, the training data collected from data providers are partly unlabeled and contain some unknown data. To address this, Lee et al. (2019); Smith et al. (2021); Kim et al. (2023); Zhu et al. (2024); Yang et al. (2024) extend Wang et al. (2021) to non-i.i.d. settings by considering the existence of out-of-distribution (OOD) data from external data. However, all of these studies focus on mitigating the negative impact of OOD data by detecting them. For example, e.g., Smith et al. (2021) treats all seen classes up to the current task as in-distribution (ID) data and uses a specific model and manually set threshold to reject OOD samples, and Kim et al. (2023) considers unknown data during inference phase and detects them by training a detector head.

Instead of simply detecting and eliminating the unlabeled data with unknown classes, can we better leverage these data for continual learning? We rethink the problem from the open-world perspective: although these data belong to novel classes that are different from seen CL task classes up to the current task, some of these classes may become future task classes in CL, e.g., an unseen class "car" at the current task might be included in upcoming CL tasks. In this case, open-world unlabeled data can help mitigate distribution shifts between different CL tasks if we can exploit the patterns within unlabeled data, especially those of novel classes. Motivated by this, this paper investigates a new question in semi-supervised CL:

*Instead of identifying and rejecting unknown classes in unlabeled data, can we fully leverage open-world data to adapt a model to new tasks and improve the overall performance in CL?*

To answer this question, we consider open semi-supervised continual learning (Open SSCL) where unlabeled datasets not only include *seen classes* up to the current CL task but also *unseen classes* from the upcoming tasks and *unknown classes* that are not part of the CL task stream. Compared to previous semi-supervised CL settings, it considers a more generalized unlabeled dataset where samples are from both CL task classes and unexpected unknown classes without task identifiers. Moreover, Open SSCL poses a unique challenge of determining which samples are relevant to the CL task stream and how to utilize those valuable samples to make the model less sensitive to distribution shifts between tasks. The goal in Open SSCL is to continuously learn a model from both labeled and unlabeled data in an open-world environment without forgetting, and meanwhile effectively utilizing unlabeled data to adapt to novel classes. In other words, we aim to use easy-to-obtain unlabeled open-world data to improve model performance on past, current, and future tasks.

Toward this end, we propose an **Open** semi-supervised learning framework **A**dapting the model to new tasks in **C**ontinual **L**earning (OpenACL). OpenACL learns unique proxies as representative embeddings to capture characteristics of data belonging to both seen and novel classes. It actively prepares the model for the CL task stream by learning the generalized representation function from unlabeled data and adapting these proxies for upcoming tasks. Additionally, learning from seen classes within the unlabeled data helps the model reinforce its memory of previously learned tasks, thereby mitigating catastrophic forgetting. Our contributions can be summarized as follows:

- We formulate a problem of open semi-supervised continual learning (Open SSCL). It is motivated by the fact that real data in practice mostly contains limited labeled data and large-scale unlabeled data, with the existence of novel classes in unlabeled data. Notably, instead of identifying and rejecting data from novel classes, Open SSCL utilizes them to enhance performance on new tasks.

- We propose OpenACL to solve the Open SSCL problem. It maintains multiple proxies for seen tasks and reserves extra proxies for unseen tasks. By learning from both labeled and unlabeled data, our method tackles catastrophic forgetting by establishing proxies for seen classes while simultaneously improving the adaptation ability by actively linking reserved proxies with classes in new CL tasks.

- We conduct extensive experiments to evaluate OpenACL and study the impact of using unlabeled data in CL. We also extend the existing CL methods to the Open SSCL setting and compare them with ours under a fair environment. The online continual learning results show that OpenACL consistently outperforms others in adapting to new tasks and addressing catastrophic forgetting.

## 2 Related Work

This paper is closely related to the literature on continual learning, semi-supervised learning, and open set/world problems. We introduce each topic and discuss the differences with our work below.

**Continual Learning (CL).** The goal is to learn a model continuously from a sequence of tasks (non-stationary data). One of the challenges in CL is to overcome the issue of catastrophic forgetting, i.e., prevent forgetting the old knowledge as the model is learned on new tasks. Various approaches have been proposed to prevent catastrophic forgetting, including regularization-based methods, rehearsal-based methods, parameter isolation-based methods, etc. Specifically, *regularization-based* methods prevent forgetting the old knowledge by regularizing model parameters; examples include Elastic Weight Consolidation (Kirkpatrick et al., 2017), Synaptic Intelligence (Zenke et al., 2017), Incremental Moment Matching (Lee et al., 2017), etc. In contrast, *rehearsal-based* methods (Rebuffi et al., 2017; Lopez-Paz & Ranzato, 2017; Saha et al., 2021) tackle the problem by reusing the old data (stored in a memory-efficient replay buffer) in previous tasks during the training process. Unlike these approaches where a single model is used for all tasks, *parameter isolation-based* methods (Mallya & Lazebnik, 2018) aims to improve the model performance on all tasks by isolating parameters for specific tasks. Note that all the above methods were studied in the classic supervised learning setting. In contrast, our paper considers an open semi-supervised setting with not only labeled data but also unlabeled data that is possibly from unknown classes.

**Semi-Supervised Learning (SSL).** It aims to learn a model from both labeled and unlabeled data, and the labeled data are usually limited while the unlabeled ones are sufficient. *Pseudo-labeling-based methods*, as discussed by Xie et al. (2020); Xu et al. (2021); Sohn et al. (2020), initially train models using labeled data and subsequently assign pseudo labels to the unlabeled data, and utilize these sample-pseudo-label pairs to further improve the model. On the other hand, *consistency regularization-based* methods (Sajjadi et al., 2016; Meel & Vishwakarma, 2021) learn to ensure consistency across different data. They augment the unlabeled data in different views of data (e.g., by rotation, scaling, etc.), and a model is then trained on the augmented data via regularized optimization such that the predictions for different views are consistent. While SSL has shown success in many tasks, its application to CL is less studied. Because unlabeled data in practice may not follow the identical distribution as the labeled data and they may come from different classes, SSL methods introduced above may not perform well in real applications. This paper closes the gap where we focus on CL and extend SSL to the open setting.

**Open-Set & Open-World Recognition.** It considers scenarios where the data observed during model deployment may come from unknown classes that do not exist during training. The goal is to not only accurately classify the seen classes, but also effectively deal with unseen ones, e.g., either distinguish them from the seen classes (open-set problem) or label them into new classes (open-world problem). The existing methods for open-set recognition include traditional machine learning-based methods (Bendale & Boult, 2015; Mendes Júnior et al., 2017; Rudd et al., 2017) and deep learning-based methods (Dhamija et al., 2018; Shih et al., 2019; Yu et al., 2017; Yang et al., 2019). OOD detection problem has also been discussed in CL tasks (Kim et al., 2022b;a). However, we consider open settings but primarily focus on semi-supervised continual learning, where the model is trained from a sequence of tasks and the training dataset includes both labeled and unlabeled data.

**Open-Set/World Semi-Supervised Learning.** It combines both open-set/world recognition and SSL. The goal is to train a model from both labeled and unlabeled data, where the unlabeled data may contain novel classes. One of the challenges is to make SSL less vulnerable to novel classes as they are irrelevant to labeled class training. To this end, most existing methods (Guo et al., 2020; Saito et al., 2021; Lu et al., 2022) first detect samples of novel categories, which are then rejected or re-weighted to ensure performance. For example, Guo et al. (2020) proposes a method that selectively uses unlabeled data by assigning weights to unlabeled samples. OpenMatch (Saito et al., 2021) integrates a One-Vs-All detection scheme to filter out samples from novel classes in SSL training loops. Cao et al. (2022) extends the open-set SSL and proposes open-world SSL, which requires actively discovering novel classes. It is also known as generalized category discovery (GCD) Vaze et al. (2022a). This setting is studied in (Rizve et al., 2022; Tan et al., 2023; Xiao et al., 2024) where novel classes are discovered using unlabeled sample alignment. Our paper is motivated by the idea of Open-world SSL. In particular, we note that the classes from untrained tasks in CL can indeed

be viewed as novel classes that need to be discovered, and it enables us to access open-world datasets where data may be from seen classes, unseen classes from untrained tasks, and unknown classes that are not related to CL tasks. Based on this, we study the Open SSCL problem. We will illustrate how the unlabeled data can be leveraged in Open SSCL to mitigate catastrophic forgetting and adapt a model to new tasks. Note that Open-world Continual Learning has been studied in Li et al. (2024), however, this work focuses on supervised training and testing on open-world datasets with unknown classes, which is more similar to the novel class discovery problem in CL like Roy et al. (2022); Zhou et al. (2022).

## 3 Problem Formulation

In this section, we formulate the problem of open semi-supervised continual learning (Open SSCL). Consider a CL problem that aims to learn a model from a sequence of $k$ tasks $T = \{T_1, \cdots, T_k\}$. Let $\mathcal{D} = \{\mathcal{D}_l, \mathcal{D}_u\}$ be a dataset associated with these tasks; it consists of $n$ labeled data samples $\mathcal{D}_l = \{(x_i, y_i)\}_{i=1}^n$ and $m$ unlabeled samples $\mathcal{D}_u = \{x_i\}_{i=1}^m$, where $m \gg n$, feature $x_i \in \mathcal{X}$, and label $y_i \in \mathcal{Y} = \{1, \cdots, N\}$. Under this semi-supervised continual learning, $\mathcal{D}_l$ is divided into multiple task sets $\mathcal{D}_l = \cup_{i \in \{1, \cdots, k\}} \mathcal{D}_l^i$ based on labels (e.g., dividing CIFAR-10 dataset into 5 tasks with two labels for each task). For each task $T_i$, we can only access labeled samples from a subset $\mathcal{D}_l^i \subset \mathcal{D}_l$ and unlabeled samples from $\mathcal{D}_u$.

We shall consider semi-supervised continual learning in an open environment, where unlabeled data $x \in \mathcal{D}_u$ may come from the known classes $C_l$ in labeled dataset $\mathcal{D}_l$ or unknown classes $C_n$, i.e., unlabeled data $\mathcal{D}_u$ is from classes $C_u = C_l \cup C_n$. In the context of CL, known classes $C_l$ in $\mathcal{D}_l$ are divided into $\{C_l^1, \cdots, C_l^k\}$, with $C_l^i \cap C_l^{i+1} = \emptyset$. Because the number of learned classes is increasing along with task change in CL, we denote known classes $C_{seen}^i = \cup_{j=1}^i C_l^j$ up to task $T_i$ as the *seen classes*, the classes $C_{unseen}^i = C_l \backslash C_{seen}^i$ from future tasks as *unseen classes*, the classes $C_n$ that are not in CL tasks as *unknown classes*, and the union of unseen classes and unknown classes $C_{novel}^i = C_{unseen}^i \cup C_n$ as the *novel classes* for the task $T_i$.

The goal is to continuously learn a model $f$ from a sequence of tasks $T$ that (i) can learn from novel classes and identify them, and (ii) correctly classify known classes while avoiding forgetting the previously learned tasks as the model gets updated. To achieve this, we seek to minimize the open risk (Scheirer et al., 2014) under continual learning constraints (Lopez-Paz & Ranzato, 2017):

$$f_t = \arg\min_{f \in \mathcal{H}} \ R\left(f(\mathcal{D}_l^t)\right) + \bar{\lambda} R_{\mathcal{O}_t}(f) \tag{1}$$

$$\text{s.t.} \ R\left(f_t(\mathcal{D}_l^i)\right) \leq R\left(f_{t-1}(\mathcal{D}_l^i)\right); \forall i \in \{0, \cdots, t-1\}$$

where $R\left(f(\mathcal{D}_l^t)\right)$ denotes the empirical risk of $f$ on *known* training data at task $t$. $f_t$ is the model learned at the end of task $t$; $R_{\mathcal{O}_t}(f)$ is the *open space risk* (Scheirer et al., 2012) and is defined as

$$R_{\mathcal{O}_t}(f) = \frac{\int_{\mathcal{O}_t} f(x)\mathrm{d}x}{\int_{\mathcal{S}_t} f(x)\mathrm{d}x}.$$

where seen region $\mathcal{S}_t$ is the empirical region of high model confidence for the seen classes at task $t$:

$$\mathcal{S}_t = \left\{x \in \mathcal{X} \mid \exists y \in C_{seen}^t \text{ such that } p(y|x) \geq \tau\right\},$$

$p(y|x)$ is the predicted probability for class $y$, and $\tau$ is a confidence threshold. Open space $\mathcal{O}_t$ is the subset of $\mathcal{S}_t$ that contains novel samples (i.e., from $C_{\text{novel}}^t$) that are incorrectly classified as seen classes with high confidence:

$$\mathcal{O}_t = \left\{x \in \mathcal{D}_u \mid y \notin C_{seen}^t, \ f_t(x) \in C_{seen}^t, \ p(f_t(x)|x) \geq \tau\right\}.$$

$R_{\mathcal{O}_t}(f)$ measures the potential risk of a function $f$ misclassifying samples that are in open space $\mathcal{O}_t$. Hyperparameter $\bar{\lambda} \geq 0$ is a regularization constant. Under the constraint in equation 1, the model performance on known classes does not decrease as the model gets updated.

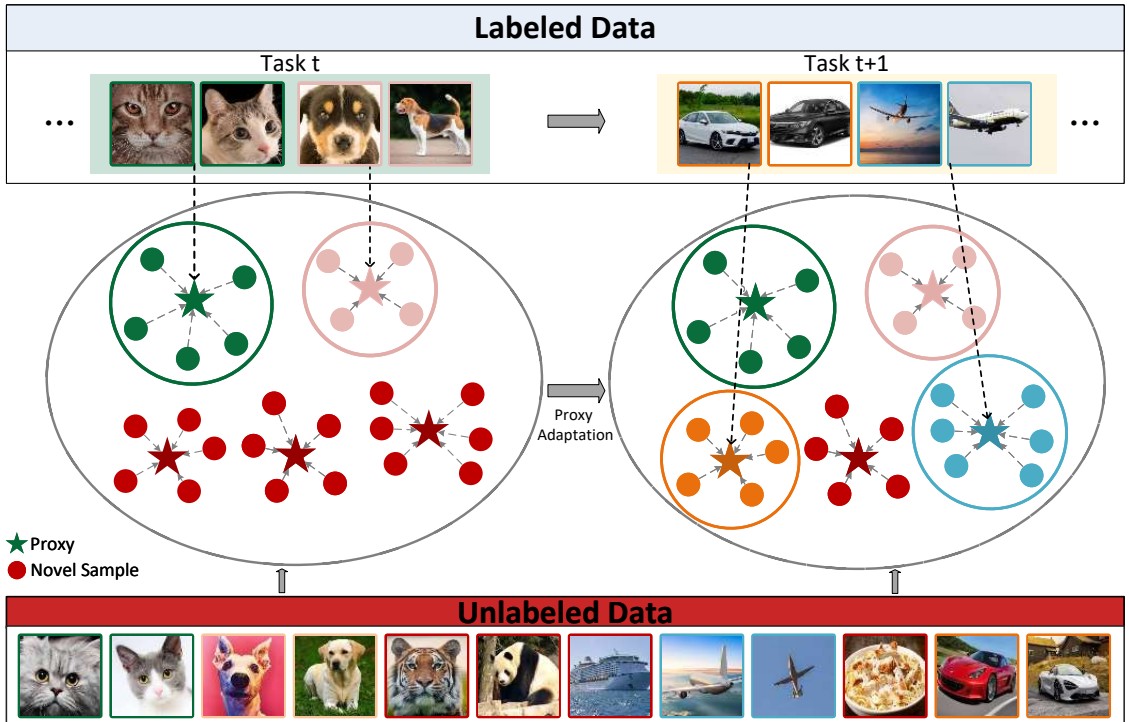

Figure 1: In OpenACL, we maintain proxies for both seen and unseen classes to learn the model from open-world data. For the continual tasks, we minimize the distance between data representations of seen classes and their associated proxies (as shown in green and pink circles). Concurrently, we leverage the semi-supervised proxy contrastive learning to learn from unlabeled data that encourages similar representations to share the same proxy distribution and learns representations for both known and novel classes by aligning their probability distribution on different proxies. We assume data from the same class have similar representations in the latent space after the representation learning, so the proxies reserved for novel classes could be used to cluster representations from novel classes, and these novel classes might be related to future continual learning tasks. Therefore, we consider an active adaptation strategy to dynamically associate proxies with future tasks. Specifically, we introduce an adaptation phase upon entering a new task $t+1$, where we receive labeled data in task $t+1$. For each class within the task, we identify the proxy from the proxy pool (red pentagrams) that is the most similar to representations of the class and allocate the class label to the proxy. By assigning novel proxies to incoming new task classes, we could have some well-trained proxies and speed up the learning process for the new task.

## 4 Proposed Method

The key challenge in Open SSCL is to exploit unlabeled open-world datasets to simultaneously solve catastrophic forgetting and improve the adaptation ability on new tasks of continual models. In this section, we introduce a novel method *OpenACL* for Open SSCL to learn from CL tasks and unlabeled open-world datasets. Instead of directly learning from representations, OpenACL learns proxies as representative embeddings that fit the centers of representations to characterize each class in the latent space.

**Using proxies to characterize each class.** In supervised learning, it is straightforward to characterize a class (distribution) by averaging image representations as the geometric center of a class. However, for novel classes, the absence of labels makes it unclear which data points should be averaged to find the representation centers. Therefore, we consider using *trainable parameters* as proxies to estimate the distributions of each class without requiring explicit labels. Specifically, we call the class proxies associated with seen classes the "seen proxies", and the proxies reserved for potential future classes are termed "novel proxies". These novel proxies are trained to capture the patterns of classes in future tasks even before they are officially labeled, enabling OpenACL to anticipate and quickly adapt to new tasks as they are introduced. Formally, we define

the set of proxies as $\mathcal{G} = \{g_1 \ldots g_{|C_l \cup C_n|}\}$ where $|C_l \cup C_n|$ represents the total number of seen classes $C_l$ and novel classes $C_n$.[2]

In particular, OpenACL updates proxies using both labeled and unlabeled data to continually learn from the task stream and enhance model robustness against distribution shift while strengthening its memory of previously seen tasks. In addition, a proxy adaption method is introduced to identify and allocate the most relevant novel proxies to incoming task classes during task transitions, thereby facilitating rapid adaptation to new tasks. We introduce the framework of OpenACL in Figure 1 and the algorithm in Algorithm 1.

### 4.1 Proxy Learning on Labeled Data

In the context of labeled data, our objective is to learn *proxies* that closely align with the seen class representations and make predictions. We achieve this by minimizing the distance between each data representation and its corresponding class proxy. Formally, given a labeled dataset $\mathcal{D}_l = \{(x_i, y_i)\}_{i=1}^n$ where the ground truth of data $x_i$ is known, we aim to maximize the cosine similarity $\text{sim}(g_{y_i}, h(x_i)) = \frac{g_{y_i}^T h(x_i)}{||g_{y_i}|| \cdot ||h(x_i)||}$ between the data representation $h(x_i)$ and its class proxy $g_{y_i}$, where $h$ is a representation function. We thus define the loss function $\mathcal{L}_p$ that encourages the data to be closer to its class proxy at task $t$ as:

$$\mathcal{L}_p = -\frac{1}{|B_l|} \sum_{i=1}^{|B_l|} \log \frac{\exp\left(\text{sim}\left(g_{y_i}, h(x_i)\right) \times s\right)}{\sum_{j=1}^{|\mathcal{G}|} \exp\left(\text{sim}\left(g_j, h(x_i)\right) \times s\right)} \tag{2}$$

In equation 2, $|B_l|$ is the number of labeled samples in a batch $B_l$. The parameter $s$ controls the softmax temperature when transforming similarity into probability, ensuring stable training (Wang et al., 2018) by enhancing the model's sensitivity to differences in similarity. By minimizing $\mathcal{L}_p$, we align representations with their corresponding class proxies while distancing them from proxies of other classes, and the label information helps to build a solid mapping from each proxy to its associated seen task class. Additionally, similar to other continual learning approaches, we utilize a replay memory mechanism to store labeled data. At each training iteration, the stored labeled samples are updated in replay memory using the loss function described in equation 2.

### 4.2 Semi-Supervised Proxy Representation Learning

To equip the model with the ability to exploit the open-world data and represent novel classes, we introduce semi-supervised proxy contrastive learning to learn robust and discriminative representations for both unlabeled and labeled data by assigning data with similar representations to a common proxy, thereby capturing the underlying class structures—even for novel classes that the model has not encountered before. Contrastive learning is designed to extract meaningful representations by exploiting both the similarities and dissimilarities between data instances. This is typically achieved by comparing two augmented views (e.g., rotation, flipping, resizing) of the same instance or different instances. However, in the context of open-world continual learning, solely focusing on instance-level alignment is insufficient for capturing the semantic structures of novel classes within unlabeled data. Instead, our objective shifts toward maintaining consistency in the distribution of representations over a set of trainable proxies.

Given an instance $x$, we generate two augmented views $\tilde{x}$ and $\tilde{x}'$ and obtain their representations $h(\tilde{x})$ and $h(\tilde{x})'$ as suggested in (Chen et al., 2020). The probability of a view $\tilde{x}$ being assigned to a proxy $g_i$ can be computed as:

$$p_i(\tilde{x}) = \frac{\exp\left(\text{sim}\left(g_i, h(\tilde{x})\right) \times s\right)}{\sum_{j=1}^{|\mathcal{G}|} \exp\left(\text{sim}\left(g_j, h(\tilde{x})\right) \times s\right)} \tag{3}$$

---

[2]It's worth noting that the number of proxies is not necessarily the same as the total number of seen and novel classes. Indeed, our method doesn't require us to know the number of novel classes in advance and the number of proxies can be flexible and dynamically adjusted as needed. We explore the impact of varying the number of proxies and propose how to dynamically increase them in Ablation Studies 5.3.

To align the distribution of novel classes over proxies between two views via optimizing the following:

$$\mathcal{L}_c^u = -\frac{1}{|\mathcal{B}_u|} \sum_{i=1}^{|\mathcal{B}_u|} \log \frac{\exp(\text{sim}(p(\tilde{x}_i), p(\tilde{x}_i'))/\kappa)}{\sum_{j=1}^{|\mathcal{B}_u|} \mathbf{1}_{[j \neq i]} \exp\left(\text{sim}(p(\tilde{x}_i), p(\tilde{x}_j))/\kappa\right)} \tag{4}$$

where $\mathcal{B}_u$ is an unlabeled minibatch including pairs of two augmented views $\tilde{x}$ and $\tilde{x}'$ from $x$. $\kappa$ is a temperature parameter where a smaller value of $\kappa$ produces a sharper probability distribution, making the model sensitive to small variations in similarities between $p(\tilde{x}_i)$ and $p(\tilde{x}_i')$. $\mathbf{1}_{[\cdot]} \in \{0, 1\}$ is the condition function. Here, labeled data could also be incorporated to improve the robustness of representations, thus, we also leverage labeled data to extend the unsupervised proxy contrastive learning to semi-supervised proxy contrastive learning. This is advantageous as the labeled data can provide direct information about the relationship between instances and their corresponding proxies. Following Khosla et al. (2020), we incorporate supervised signal in our proxy representation learning. For samples in labeled minibatch $B_l$, we also augment them in another view and build a minibatch $\mathcal{B}_l$, then we have a conjunct contrastive loss on proxy distribution:

$$\mathcal{L}_c = \mathcal{L}_c^u - \sum_{i=1}^{|\mathcal{B}_l|} \log \frac{1}{|P_i|} \sum_{\tilde{x}_j \in P_i} \frac{\exp(\text{sim}(p(\tilde{x}_i), p(\tilde{x}_j))/\kappa)}{\sum_{\tilde{x}_k \in \mathcal{B}_l \setminus \{\tilde{x}_i\}} \exp\left(\text{sim}(p(\tilde{x}_i), p(\tilde{x}_k))/\kappa\right)} \tag{5}$$

Here, $P_i$ is the set of all positive samples $\{\tilde{x}_j \in \mathcal{B}_l \setminus \{\tilde{x}_i\} : y_j = y_i\}$.

The final objective combines the proxy contrastive loss and the supervised loss, weighted by a hyper-parameter $\lambda$, i.e., the loss at task $t$ is $\mathcal{L} = \mathcal{L}_p + \lambda \mathcal{L}_c$. We set $\lambda$ as 1 in our method.

The rationale of the proxy-level contrastive learning mechanism is multifold. By aligning the proxy distributions of augmented views, we encourage instances with similar semantic content to be associated with the same proxies. These proxies act as class-level anchors, and aligning distributions to these proxies naturally suppresses overconfident predictions on novel data that would otherwise fall into the open space $\mathcal{O}_t$. Specifically, we reserve redundant proxies for unknown samples to encourage novel or out-of-distribution data to be mapped distinctly, instead of being falsely assigned to high-confidence regions associated with seen classes. We also align labeled samples with their proxies to compact the region $\mathcal{S}_t$ by increasing classification confidence. As a result, our method actively reduces the probability mass assigned to novel samples in high-confidence regions of seen classes, thereby minimizing $R_{\mathcal{O}_t}(f)$ as formalized in equation 1. Furthermore, since the unlabeled data in equation 5 may include samples from previously trained tasks, the current task, and future tasks, our model leverages a comprehensive proxy representation that spans the entire task continuum. This inherently provides a regularizing effect to make up for catastrophic forgetting, minimizing the risk of overriding previous information.

### 4.3 Continual Proxy Adaptation for New Tasks

The aforementioned proxy learning establishes a set of novel proxies learned from the intra-class similarities within the unlabeled data. As new task classes may related to unlabeled data, we can further leverage these novel proxies to adapt the CL model to a new task. Intuitively, upon transitioning from task $t$ to task $t + 1$, the classes in the forthcoming task should already possess associated proxies, considering their presence in the unlabeled data and our proxy representation learning group similar unlabeled data. Thus, we could associate potential proxies with new classes shown in the new task $t + 1$ and adapt the model to the task quickly.

Specifically, consider labeled data $\{(x, y) \in \mathcal{D}_l^{t+1}\}$ at new task $t + 1$. For each class label $\bar{y} \in C_l^{t+1}$, we want to find the most potential proxy for class $\bar{y}$ by measuring the similarity of proxies to samples from class $\bar{y}$. Define a count function $I(x, g_j)$ that returns 1 if $g_j$ is the most similar proxy for $x$ and 0 otherwise:

$$I(x, g_j) = \begin{cases} 1 & \text{if } g_j = \arg\max_{g_k \in \mathcal{G}} \text{sim}(x, g_k) \\ 0 & \text{otherwise} \end{cases} \tag{6}$$

We then determine the proxy $g_{\bar{y}}^* \in \mathcal{G}$ by the number of its closet samples in $\{(x,y) \in \mathcal{D}_l^{t+1} : y = \bar{y}\}$. The one with the most grouped samples will be selected as $g_{\bar{y}}^*$ and be assigned with label $\bar{y}$.

$$g_{\bar{y}}^* = \arg \max_{g_j \in \mathcal{G}} \sum_{(x_i,y_i) \in \mathcal{D}_l^{t+1}:y_i=\bar{y}} I(x_i, g_j) \tag{7}$$

In the implementation, if multiple classes in the new task $t+1$ are associated with the same proxy, we randomly assign a class label $y$ from these classes to the proxy. In addition, to avoid the trivial solution that all unlabeled instances are assigned to a single proxy in the early stage of the training (Caron et al., 2018; Cao et al., 2022), we adopt a reinitialization strategy. After assigning labels for task $t+1$ but before entering its training, the unassigned novel proxies are reinitialized. To establish these initial novel proxies as a new task begins, we deploy the $K$-means algorithm, using cosine distance as a metric to cluster centroids as initial novel proxies. The known proxies are used as prior knowledge for the $K$-means algorithm, but remain static and are not subjected to updates post-clustering. Specifically, given the proxy pool $\mathcal{G}$ and the set of seen class proxies for $C_s^{t+1}$, the initialized centroids in $K$-means algorithm are selected as $|C_s^{t+1}|$ known proxies and $|\mathcal{G}| - |C_s^{t+1}|$ randomly selected data points from the unlabeled dataset $D_u$. To reduce computation cost, $K$-means is running on a subset of $D_u$ to obtain $|\mathcal{G}|$ centroids. we identify $|C_s^{t+1}|$ centroids that are most similar to the known proxies and exclude them using cosine similarity. The remaining centroids are used to initialize the novel proxies in the proxy pool. This ensures a more representative set of proxies for subsequent tasks and solves the trivial solution problem.

---

**Algorithm 1** OpenACL

**Require:** tasks $T_1, ..., T_k$; labeled dataset $\mathcal{D}_l$ and unlabeled dataset $\mathcal{D}_u$; memory $\mathcal{M}$; proxies $\mathcal{G}$; representation function $h$; task classes $C_l = \{C_l^1, ..., C_l^k\}$; temperature parameters $s$ and $\kappa$; learning rate $\eta$

1: **for** $t \in \{T_1, \ldots, T_k\}$ **do**
2:      **if** $t \neq T_1$ **then**
3:          **for** $\bar{y} \in C_l^t$ **do**
4:              $g_{\bar{y}} = \arg \max_{g_j \in \mathcal{G}:j \geq \sum_{i=1}^{t-1} |C_l^i|} \sum_{(x_i,y_i) \in \mathcal{D}_l^{t+1}:y_i=\bar{y}} I(x_i, g_j)$          ▷ Proxy Adaptation
5:          **end for**
6:          **for** $j = \max(C_l^t) + 1$ to $m$ **do**
7:              $g_j = \text{reinitialize}(\mathcal{D}_u)$
8:          **end for**
9:      **end if**
10:      **for** a batch $B_l = \{(\tilde{x}_i, \tilde{x}_i', y_i)\}_{i=1}^{|B_u|} \subset \mathcal{D}_l^t$ **do**
11:          $B_u = \{(\tilde{x_i^u}, \tilde{x_i^u}')\}_{i=1}^{|B_u|} \subset \mathcal{D}_u$          ▷ Random Sample from $\mathcal{D}_u$
12:          $\mathcal{L}_p = -\frac{1}{|B_l|} \sum_{i=1}^{|B_l|} \log \frac{\exp\left(\text{sim}\left(g_{y_i}, h(\tilde{x}_i)\right) \times s\right)}{\sum_{j=1}^{|\mathcal{G}|} \exp\left(\text{sim}(g_j, h(\tilde{x}_i)) \times s\right)}$
13:          $\mathcal{L}_c^u = -\frac{1}{|\mathcal{B}_u|} \sum_{i=1}^{|\mathcal{B}_u|} \log \frac{\exp(\text{sim}(p(\tilde{x_i^u}), p(\tilde{x_i^u}'))/\kappa)}{\sum_{j=1}^{|\mathcal{B}_u|} \mathbf{1}_{[x_j \neq x_i]} \exp\left(\text{sim}(p(\tilde{x_i^u}), p(\tilde{x_j^u}))/\kappa\right)}$
14:          $\mathcal{L}_c = \mathcal{L}_c^u - \sum_{i=1}^{|\mathcal{B}_l|} \log \frac{1}{|P_i|} \sum_{\tilde{x}_j \in P_i} \frac{\exp(\text{sim}(p(\tilde{x}_i), p(\tilde{x}_j))/\kappa)}{\sum_{\tilde{x}_k \in \mathcal{B}_l \setminus \{\tilde{x}_i\}} \exp\left(\text{sim}(p(\tilde{x}_i), p(\tilde{x}_k))/\kappa\right)}$
15:          $\mathcal{G}, h = \text{GradientDescent}(\mathcal{L}_p + \mathcal{L}_c; \mathcal{G}, h, \eta)$
16:          $\mathcal{M} = \text{Update}(\mathcal{M}, B_l)$
17:          $\mathcal{G}, h = \text{MemoryReplay}(\mathcal{M}; \mathcal{G}, h, \eta)$
18:      **end for**
19: **end for**
20: **Output** $\mathcal{G}$ and $h$

---

## 5 Experiments

In this section, we introduce the datasets and the baselines. Then, we present results from various benchmarks in comparison to baselines. Implementation details are available in Appendix B.1.

### 5.1 Experiment Setting

**Datasets.** We adopt the following datasets in experiments. The data from known classes is partitioned into labeled and unlabeled segments with ratios of 20% labeled data and 50% labeled data. We also examine more extreme scenarios in Appendix C.2 to evaluate the model with 10% and 80% labeled data ratios.

1. **CIFAR-10 (Krizhevsky et al., 2009):** The first 6 classes are organized into 3 tasks ($k = 3$), each containing two classes. The remaining 4 classes are treated as unknown. For each task, we have 2,000 labeled instances under the 20% split and 5,000 labeled instances under the 50% split.

2. **CIFAR-100 (Krizhevsky et al., 2009):** The initial 80 classes from CIFAR-100 are segmented into 16 tasks ($k = 16$). The subsequent 20 classes are treated as unknown. For every task, 500 instances are labeled under the 20% split, and 1,250 instances are labeled under the 50% split.

3. **Tiny-ImageNet (Deng et al., 2009; Le & Yang, 2015):** The initial 120 classes of Tiny-ImageNet are divided into 20 tasks ($k = 20$), leaving 80 classes as unknown. For each task, there are 600 labeled instances in the 20% split and 1,500 labeled instances in the 50% split.

Using the above split, we take two datasets as input: labeled $D_l = \{\mathcal{D}_l^1, ..., \mathcal{D}_l^k\}$ and unlabeled $D_u$ consisting of unlabeled data from **known classes** $C_l$ and all data from **unknown classes** $C_n$. For each task $i$, we simultaneously sample data from the $\mathcal{D}_l^i$ for the current task and the $D_u$. The proportion of labeled to unlabeled data in the sample matches the respective proportions in the datasets. Note that, we consider $D_u$ is from open-world, so it covers all classes. Therefore, $D_u$ and $D_l$ come from two different distributions. We sequentially sampled the data from the $D_u$ without knowing the source, i.e., the data comes from previous task classes, current task classes, future task classes, or unknown classes. Datasets are introduced in detail in Appendix B.3. In addition to these datasets, we also evaluate our method on a naturally-shifted dataset: Stanford Cars. The results are provided in Appendix C.3.

**Baselines.** We compare OpenACL with existing methods in CL in both *task incremental learning (Task-IL)* and *class incremental learning (Class-IL)* settings. The distinction between these settings is elaborated upon in Appendix B.1. Additionally, our focus is on online continual learning, where models are only allowed to be trained for 1 epoch. However, we still give the results for multiple epoch training in Appendix C.1. To ensure a fair comparison, we first equip supervised learning-based methods with a well-known SSL method: FixMatch (Sohn et al., 2020). Unlabeled samples with low prediction confidence would be rejected during train and only those with high confidence would be pseudo-labeled. Then, as our method is adapted from the contrastive learning idea to align the distribution, we also add a contrastive learning loss (Chen et al., 2020) to baselines to learn representation from unlabeled data. These baselines include: *Joint*, *Independent* (Lopez-Paz & Ranzato, 2017), *GEM* (Lopez-Paz & Ranzato, 2017), *iCaRL* (Rebuffi et al., 2017), *GSS* (Aljundi et al., 2019), *ER* (Chaudhry et al., 2019), *DER* (Buzzega et al., 2020), *ER-ACE* (Caccia et al., 2022), *DER* (Buzzega et al., 2020), *ER-ACE* (Caccia et al., 2022), *DVC* (Gu et al., 2022), *DistillMatch* (Smith et al., 2021), *AutoNovel* (Han et al., 2020), *FACT* (Zhou et al., 2022), *ORCA* (Cao et al., 2022), *Refresh* (Wang et al., 2024), and VR-MCL (Wu et al., 2024). We introduce these baselines in Appendix B.4.

### 5.2 Results

**Evaluation on split datasets.** We contrasted our algorithm against established baselines in the online Task-IL setting and online Class-IL setting with varying label ratios across seen classes. To make a fair comparison, supervised continual learning methods are integrated with FixMatch or SimCLR. Table 1 and 2 present the mean accuracy across all tasks for each method, both with and without the inclusion of unlabeled data. The results in the Task-IL setting and Class-IL setting demonstrate that OpenACL shows better performance compared with baselines. When there are more classes in the data, the advantage becomes more obvious. Notably, we observe that some baselines also benefit from unlabeled data enhanced by FixMatch or SimCLR. This emphasizes the potential benefits of unlabeled data in the context of CL. However, directly integrating CL with unlabeled data usage yields only modest improvements, highlighting the need for more specialized methods for Open SSCL, like OpenACL. OpenACL's superior performance suggests that specialized algorithms tailored for Open SSCL can provide considerable benefits over traditional methods or straightforward combinations of the existing methods.

Table 1: Average accuracy over three runs of experiments on Task-IL benchmarks. Some baselines are adapted to SSL by incorporating them with FixMatch (Sohn et al., 2020) or SimCLR (Chen et al., 2020) to learn from unlabeled data. Results are organized as SimCLR usage / FixMatch usage / No unlabeled data usage. The standard deviation results are reported in the Appendix D.

| Method | CIFAR-10 | | CIFAR-100 | | Tiny-ImageNet | |
|---|---|---|---|---|---|---|
| Labels % | 20 | 50 | 20 | 50 | 20 | 50 |
| Joint | 68.3 / 68.9 / 67.9 | 69.1 / 69.4 / 68.7 | 68.4 / 68.1 / 67.5 | 76.6 / 75.7 / 75.1 | 52.8 / 50.3 / 50.7 | 58.3 / 57.8 / 57.0 |
| Single | 57.5 / 57.6 / 54.7 | 59.3 / 57.0 / 57.6 | 33.5 / 34.1 / 32.3 | 37.9 / 36.3 / 37.2 | 20.9 / 20.5 / 19.6 | 25.9 / 23.3 / 23.1 |
| Independent | 62.5 / 64.2 / 61.3 | 63.9 / 62.3 / 62.5 | 26.7 / 30.3 / 31.8 | 36.2 / 36.2 / 33.4 | 21.6 / 21.5 / 23.2 | 26.5 / 28.0 / 27.0 |
| iCaRL | 56.0 / 57.4 / 56.7 | 57.2 / 58.7 / 58.3 | 45.8 / 45.9 / 46.4 | 44.1 / 42.3 / 41.8 | 25.2 / 25.3 / 23.5 | 31.3 / 29.0 / 26.5 |
| DER | 62.2 / 63.9 / 63.3 | 63.2 / 63.2 / 63.6 | 38.6 / 38.7 / 39.6 | 46.8 / 44.7 / 44.0 | 24.2 / 22.4 / 25.8 | 28.4 / 29.6 / 28.0 |
| GEM | 61.3 / 64.0 / 62.6 | 63.2 / 63.6 / 64.2 | 53.5 / 52.6 / 51.8 | 58.6 / 57.5 / 54.4 | 33.0 / 35.4 / 32.1 | 40.1 / 37.3 / 38.0 |
| ER | 62.9 / 62.3 / 61.3 | 64.9 / 63.8 / 62.6 | 54.8 / 55.3 / 53.7 | 59.9 / 58.5 / 57.8 | 35.2 / 36.3 / 35.7 | 41.7 / 41.4 / 40.2 |
| ER-ACE | 61.2 / 61.6 / 61.3 | 62.4 / 64.2 / 63.9 | 53.8 / 55.0 / 54.8 | 61.7 / 62.4 / 62.1 | 36.2 / 37.2 / 35.4 | 41.4 / 42.4 / 40.6 |
| Refresh | 63.0 / 63.1 / 61.7 | 62.6 / 64.3 / 62.6 | 54.7 / 55.3 / 55.1 | 61.2 / 61.9 / 61.0 | 35.8 / 36.9 / 35.8 | 42.6 / 42.2 / 41.5 |
| VR-MCL | 60.3 | 63.4 | 53.3 | 63.3 | 32.9 | 40.3 |
| DVC | 57.4 | 61.7 | 57.6 | 62.7 | 36.8 | 43.5 |
| DistillMatch | 57.8 | 59.4 | 35.7 | 41.3 | 21.8 | 26.2 |
| AutoNovel | 56.3 | 56.5 | 58.7 | 63.3 | 37.4 | 43.1 |
| FACT | 53.2 | 55.3 | 55.9 | 62.8 | 35.0 | 42.3 |
| ORCA | 60.9 | 62.2 | 56.4 | 62.4 | 34.4 | 39.3 |
| **OpenACL** | **64.3** | **66.3** | **60.4** | **66.6** | **40.2** | **47.0** |

Table 2: Average accuracy over three runs of experiments on Class-IL benchmarks.

| Method | CIFAR-100 | | Tiny-ImageNet | |
|---|---|---|---|---|
| Labels % | 20 | 50 | 20 | 50 |
| Joint | 22.8 / 23.0 / 21.8 | 31.8 / 32.9 / 30.8 | 13.4 / 14.4 / 13.6 | 22.0 / 21.5 / 21.1 |
| Single | 3.1 / 2.8 / 2.5 | 3.0 / 2.5 / 3.0 | 1.9 / 2.0 / 1.7 | 2.4 / 2.8 / 2.7 |
| iCaRL | 6.8 / 7.0 / 6.3 | 7.3 / 8.3 / 7.0 | 4.5 / 3.3 / 3.4 | 4.1 / 4.8 / 4.2 |
| DER | 3.7 / 3.7 / 3.5 | 3.6 / 3.9 / 3.9 | 2.4 / 2.5 / 2.1 | 2.4 / 2.6 / 2.3 |
| GEM | 7.0 / 8.0 / 6.9 | 9.7 / 7.7 / 6.7 | 2.4 / 3.4 / 2.7 | 2.3 / 2.6 / 1.8 |
| GSS | 12.8 / 11.2 / 10.3 | 16.8 / 15.3 / 15.2 | 3.3 / 5.4 / 3.8 | 5.3 / 5.6 / 5.0 |
| ER | 10.9 / 12.0 / 11.5 | 15.6 / 15.8 / 16.9 | 3.3 / 4.2 / 3.9 | 4.8 / 6.7 / 5.7 |
| ER-ACE | 12.8 / 13.3 / 12.0 | 16.7 / 17.9 / 17.1 | 5.0 / 5.4 / 4.9 | 7.4 / 8.1 / 7.2 |
| Refresh | 10.6 / 11.6 / 11.2 | 16.9 / 18.1 / 17.3 | 5.2 / 5.5 / 5.4 | 6.6 / 7.3 / 6.9 |
| VR-MCL | 12.3 | 17.4 | 5.2 | 7.5 |
| DVC | 11.2 | 16.2 | 5.8 | 8.3 |
| DistillMatch | 2.8 | 3.2 | 2.0 | 2.7 |
| AutoNovel | 13.2 | 17.9 | 6.5 | 9.2 |
| FACT | 12.9 | 16.3 | 5.9 | 8.2 |
| ORCA | 14.4 | 18.8 | 6.8 | 9.6 |
| **OpenACL** | **15.7** | **20.0** | **7.9** | **11.9** |

Table 3: BWT and FWT results on 50% labeled dataset. We report the best results among three implementations(SimCLR, FixMatch, and Normal). The results show as BWT / FWT.

| | Single | Independent | iCaRL | DER | GEM | ER | ER-ACE | Refresh | DVC | DistillMatch | AutoNovel | FACT | ORCA | OpenACL |
|---|---|---|---|---|---|---|---|---|---|---|---|---|---|---|
| CIFAR-100 | -5.3 / 0.9 | 0 / 0 | -5.3 / 0 | 0.3 / -0.3 | 11.6 / -0.3 | 11.5 / -5.1 | 12.4 / -1.7 | **14.5** / -4.9 | 11.1 / 1.6 | -6.5 / -1.8 | 10.6 / 1.1 | 7.8 / 2.4 | 7.7 / 1.5 | 9.2 / **13.0** |
| Tiny-ImageNet | -6.3 / 0.4 | 0 / 0 | -1.1 / 0 | -0.5 / 0.8 | 4.8 / 0.1 | 4.3 / 0.6 | 6.0 / -0.1 | **6.4** / -0.2 | 5.9 / 0.5 | -11.8 / -0.1 | 4.6 / 0.9 | 3.7 / 3.9 | -0.3 / 0.2 | 2.7 / **10.9** |

**Mitigate catastrophic forgetting.** We follow Lopez-Paz & Ranzato (2017) to compare backward transfer (BWT) and forward transfer (FWT) in Table 3. Positive BTW suggests that performance on old tasks improved after learning new tasks, while a negative BWT implies that the model forgot some of the previous tasks. ER-ACE, which is a specific method for OCL achieves the best BWT among these baselines, while OpenACL achieves comparable performance as baselines on solving catastrophic forgetting. We also track the average test accuracy on the first three tasks over time to examine catastrophic forgetting. The results are presented in Figure 2. It shows that our method performs the best on the first three tasks during training and is also more stable than baselines. Besides, along with training, OpenACL even achieves better performance on the first few tasks, while some baselines almost forget the first three tasks completely, especially in challenging datasets like Tiny-ImageNet. These results validate that OpenACL can help to tackle catastrophic forgetting.

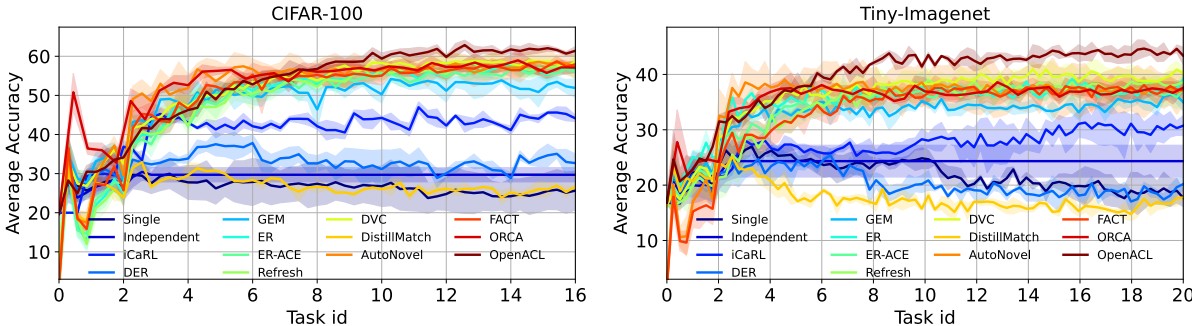

Figure 2: Average accuracy of the first three tasks on 50% labeled CIFAR-100 and Tiny-ImageNet during Task-IL training. We test the models on the first three tasks after finishing subsequent tasks to examine their ability to preserve prior knowledge.

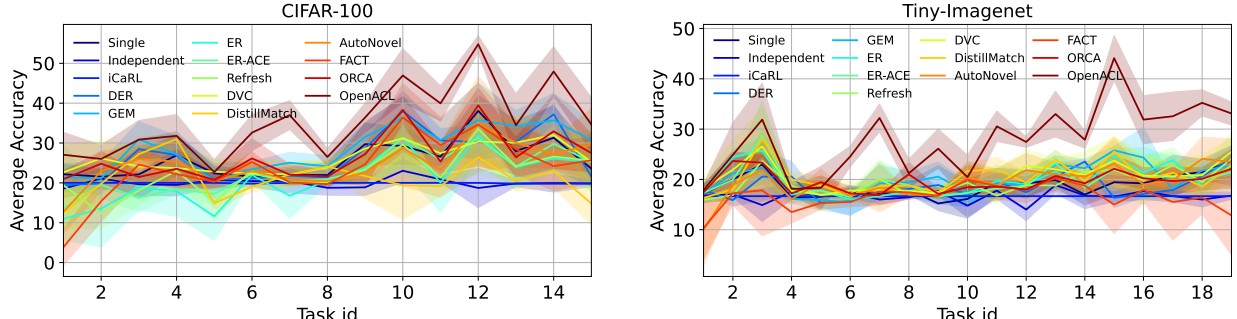

Figure 3: Average accuracy on a novel task after training with a single batch in Task-IL.

**Adaptability to new tasks.** FWT in Table 3 indicates the effect on the performance of learning new tasks from prior learning. A positive FWT suggests the model's "zero-shot" learning ability for unseen tasks. The results show that OpenACL exhibits superior performance in FWT, highlighting its exceptional zero-shot learning capability, confirming that it can swiftly adapt to new tasks leveraging unlabeled data knowledge. Further underlining its adaptability, we investigate the adaptability by comparing accuracy after training a single batch of data in a new task. Figure 3 shows OpenACL attains high accuracy across all tasks and maintains a stable performance throughout the process, suggesting that our algorithm can efficiently learn and adapt to new tasks.

Table 4: Average accuracy of the ablation study, focusing on unlabeled data usage, across three runs on CIFAR-100 and Tiny-ImageNet within the Task-IL setting.

|  | CIFAR-100 | | | Tiny-ImageNet | | |
|---|---|---|---|---|---|---|
|  | **Acc** | **BWT** | **FWT** | **Acc** | **BWT** | **FWT** |
| **OpenACL(S)** | $58.8_{\pm1.24}$ | $3.0_{\pm0.57}$ | $7.8_{\pm1.89}$ | $38.3_{\pm1.12}$ | $-0.2_{\pm0.49}$ | $4.1_{\pm0.87}$ |
| **OpenACL(N)** | $62.3_{\pm0.78}$ | $7.0_{\pm0.67}$ | $10.4_{\pm1.01}$ | $44.0_{\pm0.63}$ | $1.9_{\pm0.57}$ | $8.1_{\pm0.50}$ |
| **OpenACL** | $66.6_{\pm0.28}$ | $9.2_{\pm1.65}$ | $13.0_{\pm1.48}$ | $47.0_{\pm0.42}$ | $2.7_{\pm1.36}$ | $10.9_{\pm1.10}$ |

**Evaluation on unlabeled data.** To evaluate the impact of learning from unlabeled data in CL, we compare OpenACL with its supervised learning counterpart, OpenACL(S). OpenACL(S) conducts supervised training without the use of unlabeled data, but keeps the proxy adaptation with the $k$-means initialization. In addition to supervised learning, we examine an extreme situation in an open-world setting where unlabeled data are completely different from the CL task data. OpenACL(N) considers unlabeled data to be all from unknown classes that are entirely different from the CL task classes. The results, presented in Table 4 indicate that without the inclusion of unlabeled data during training, the performance of OpenACL(S) aligns more closely with that of ER and GEM in terms of accuracy in table 1. Although OpenACL(S) retains some zero-shot learning capabilities due to the proxy adaptation, this ability is diminished with the exclusion of unlabeled data. Notably, the results of OpenACL(N) demonstrate that even when unlabeled data consist solely of

unknown classes, they still contribute to learning the representation function and improve performance on the CL tasks. This finding suggests that the assumption requiring unlabeled data to contain potential CL task classes is not strictly necessary to effectively leverage unlabeled data in CL. By utilizing unlabeled data from unknown classes, we can still enhance the model's ability to generalize and adapt, thereby improving overall performance.

## 5.3 Ablation Studies

**Ablation Study on Adaptation:** We also conduct an ablation study on the CIFAR-100 and Tiny-ImageNet datasets by removing each component separately to examine their importance. Specifically, we systematically evaluate the impact of (i) Omitting the proxy adaptation (denoted as w/o PA), (ii) Excluding the $k$-means initialization in the proxy adaptation (denoted as w/o K), (iii) Omitting proxy allocation for new tasks while retaining the $k$-means initialization in the proxy adaptation (denoted as w/o A). The analysis of w/o PA is intended to explain the effectiveness of proxy adaptation when shifting to new tasks. Meanwhile, the evaluation of w/o K aims to affirm that the model's adaptability is mainly from our continual proxy learning mechanism, not the $k$-means initialization. OpenACL w/o A is discussed to show the sole influence of the $k$-means initialization.

Table 5: Ablation study on the proxy adaptation. We report average accuracy over three runs using different variants of OpenACL in Task-IL.

|  | CIFAR-100 | | | Tiny-ImageNet | | |
|---|---|---|---|---|---|---|
|  | **Acc** | **BWT** | **FWT** | **Acc** | **BWT** | **FWT** |
| **w/o PA** | $65.9_{\pm 0.77}$ | $10.1_{\pm 1.65}$ | $0.4_{\pm 3.40}$ | $45.6_{\pm 0.22}$ | $2.8_{\pm 0.15}$ | $0.7_{\pm 0.70}$ |
| **w/o K** | $66.2_{\pm 0.94}$ | $10.2_{\pm 1.49}$ | $9.8_{\pm 1.13}$ | $46.2_{\pm 0.36}$ | $3.4_{\pm 1.54}$ | $9.9_{\pm 0.38}$ |
| **w/o A** | $66.4_{\pm 0.38}$ | $7.6_{\pm 0.99}$ | $1.4_{\pm 1.01}$ | $45.1_{\pm 0.38}$ | $1.9_{\pm 1.01}$ | $1.0_{\pm 0.90}$ |
| **OpenACL** | $66.6_{\pm 0.28}$ | $9.2_{\pm 1.65}$ | $13.0_{\pm 1.48}$ | $47.0_{\pm 0.42}$ | $2.7_{\pm 1.36}$ | $10.9_{\pm 1.10}$ |

As shown in Table 5, the performance of OpenACL is compromised upon the removal of any single component. We mainly consider FWT in this experiment because the proxy adaptation is designed to adapt to the new tasks. A comparison between OpenACL w/o PA and OpenACL demonstrates a considerable enhancement in FWT with the use of the proxy adaptation. However, even without the proxy adaptation, the model still manages a mild positive FWT which verifies that our method can learn a general representation for both seen classes and unseen classes.

Furthermore, it also shows that the improvement of adaptation is not achieved by $k$-means initialization. By looking at OpenACL w/o K, it still achieves good performance on FWT compared with others. Therefore, $k$-means initialization is only used to amplify the adaptability of the model. Then, by analyzing the results of OpenACL w/o A, we could find that $k$-means initialization brings about a minor improvement but still serves a role in augmenting our adaptation strategy. Removing proxy allocation (w/o A) may lead to incorrect associations between representation clusters from k-means and existing class proxies. For example, representations belonging to a new class (e.g., class 0) might be incorrectly assigned to an existing proxy (e.g., proxy for class 1). This misalignment delays the model's ability to remap representations correctly and can result in noisy updates to existing class proxies, leading to a reduction in forgetting mitigation ability (decreasing in BWT). In addition, ablation on the proxy adaptation also shows this component does not markedly affect accuracy.

**Ablation Study on the Number of Proxies:** Ideally, we want the number of proxies $|\mathcal{G}|$ to match the number of all classes $|C_u|$ in the dataset. Here we evaluate using the different number of Proxies on OpenACL in Table 14. Even if $|g| \neq |C_u|$, OpenACL still attains high performance when classes don't have full support in the unlabeled data or when some proxies are not activated by data. Therefore we do not require prior knowledge of the distribution of novel classes.

Additionally, in real open-world OCL scenarios where the number of classes in the labeled dataset is unknown, the predefined proxies might be not enough during training, because we may not know the number of labeled classes at the beginning of a real-world OCL scenario. Therefore, we also study the feasibility to incrementally update the number of proxies. Here, we conduct an additional experiment (*Incremental* in Table 14) where

predefined proxies are insufficient for incoming task classes. In this experiment, we set the predefined number of proxies as 50 and 100 for CIFAR-100 (80 task classes and 20 unknown classes) and TinyImageNet (120 task classes and 80 unknown classes), respectively. If 80% proxies are assigned to task classes during training, we reinitialize another 50 proxies for CIFAR-100 and 100 proxies for Tiny-ImageNet to train the model using all proxies. The results show that OpenACL is able to dynamically increase the number of proxies, even if the predefined proxies are not enough during training (smaller than the number of labeled classes).

Table 6: Ablation study on the number of proxies.

| | CIFAR-100 | | | Tiny-ImageNet | |
|---|---|---|---|---|---|
| **Proxies** | 20 | 50 | **Proxies** | 20 | 50 |
| 90 | $60.0_{\pm 0.73}$ | $66.3_{\pm 0.15}$ | 150 | $40.4_{\pm 0.69}$ | $47.1_{\pm 0.89}$ |
| 100 | $60.4_{\pm 1.19}$ | $66.6_{\pm 0.28}$ | 200 | $40.2_{\pm 0.45}$ | $47.0_{\pm 0.42}$ |
| 200 | $60.3_{\pm 0.99}$ | $65.0_{\pm 0.68}$ | 300 | $39.7_{\pm 1.01}$ | $46.8_{\pm 0.58}$ |
| 300 | $59.6_{\pm 1.19}$ | $65.1_{\pm 0.33}$ | 400 | $39.0_{\pm 1.21}$ | $46.5_{\pm 0.75}$ |
| Incremental | $60.0_{\pm 0.99}$ | $64.9_{\pm 0.84}$ | Incremental | $40.0_{\pm 1.06}$ | $46.2_{\pm 0.81}$ |

## 6 Conclusion

In this paper, we study continual learning in an open scenario and formulate open semi-supervised continual learning. Unlike traditional CL, Open SSCL learns from both labeled and unlabeled data and allows novel classes to appear in unlabeled data. Recognizing the relationship between transitions from known tasks to upcoming tasks in CL and shifts from known classes to novel classes, we propose OpenACL. It exploits the open-world data to enhance the model's adaptability while simultaneously mitigating catastrophic forgetting. Our study highlights the importance of using unlabeled data and novel classes in CL and the potential of Open SSCL as a promising direction for future research.

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

# A  Appendix

# B  Addition Experiment Setting

## B.1  Implementation details

All experiments are conducted on a server equipped with multiple NVIDIA V100 GPUs, Intel Xeon(R) Platinum 8260 CPU, and 256GB memory. The code is implemented with Python 3.9 and PyTorch 1.10.0.

We used the same network architecture as (Lopez-Paz & Ranzato, 2017), a reduced ResNet18 for CIFAR and Tiny-ImageNet images. We consider two settings: *task incremental learning (Task-IL)* and *class incremental learning (Class-IL )*. Task-IL assumes task id is known and used to select a classifier (separate logits) for a specific task, while it is not allowed to use task id in Class-IL. Therefore the Class-IL setting is much more challenging than the Task-IL setting. Note that, OpenACL only uses the task id to separate logits for $\mathcal{L}_p$ in the Task-IL setting. In addition, the online training setting is used in our experiments where the model is only allowed to train 1 epoch on task data, every labeled and unlabeled sample is only seen once. However, we also perform 3 iterations over a batch in Class-IL following Aljundi et al. (2019). Note that, it is different from training multiple epochs on a task.

We train models using a stochastic gradient descent (SGD) optimizer. In the Task-IL setting, we allow the use of task id to separate the replay memory. The size of the replay memory is set to 250 per task under 50% labeled dataset and 125 per task under 20% labeled dataset. OpenACL uses the same memory replay strategy as the GEM to store labeled data but without the gradient projection. All baselines use the same memory replay size to make the fair comparisons. We retrieve 10 samples from the memory to replay past tasks. In the Class-IL setting, to avoid using task id, OpenACL adopts the same replay strategy as ER and uses Reservoir Sampling (Vitter, 1985) to store labeled data. For replay-based methods, the size of the replay memory is set to 4,000 and 2,000 for 50% labeled dataset and 20% labeled dataset respectively. At every iteration, we retrieve 30 samples from the replay memory. However, GEM still uses the full memory. Only equation equation 2 is used to update the model during replaying. During the training, in all experiments, we set the batch size for labeled data to 10, and the batch size of unlabeled data to $10 \cdot \frac{\mathcal{D}_u}{\mathcal{D}_l}$. It ensures that the ratio of unlabeled to labeled data in each batch is proportionate to their overall distribution in the datasets. We first shuffle the entire unlabeled dataset and then sequentially sample data unlabeled instances from it. As the ratio of labeled to unlabeled samples in each batch matches the overall ratio of the two datasets, we guarantee that each unlabeled data point is also accessed exactly once. We search and choose hyperparameters for baselines to make a fair comparison. The learning rate for baselines is searched from [0.001, 0.01, 0.05, 0.1, 0.5, 1.0] to find the best learning rate for baselines. In addition, the temperature $s$ in equation 2 and equation 3 is set to 10, as suggested in previous methods (Cao et al., 2022), and $\kappa$ is set to 0.07 as the original setting in (Chen et al., 2020). The threshold in FixMatch of baselines is set to 0.8.

## B.2  Metric

Three metrics are used in our experiments, including Accuracy (ACC), Backward Transfer (BWT), and Forward Transfer (FWT) (Lopez-Paz & Ranzato, 2017; Yan et al., 2021).

**ACC:** We report the average accuracy on all trained tasks to evaluate the fundamental classification performance of all methods.

**BWT:** BWT measures the influence of learning a new task $t$ on previous tasks $\{1, ..., t-1\}$. To calculate the BWT, we define accuracy on test classes $C_l^t$ at task $t$ as the $A_{C_l^t}^t$. BWT is computed as follows:

$$\text{BWT} = \frac{1}{|T-1|} \sum_{i=2}^{|T|} \frac{1}{i} \sum_{j=1}^{i} A_{C_l^i}^i - A_{C_l^j}^j \tag{8}$$

**FWT:** FWT gauges how the model performs on upcoming task $t + 1$ at task $t$. Let $\bar{a}$ be a vector storing accuracy for all tasks at random initialization status. After finishing all the tasks, we have FWT:

$$\text{FWT} = \frac{1}{|T - 1|} \sum_{i=2}^{|T|} A_{C_l^i}^{i-1} - \bar{a}_i \tag{9}$$

### B.3 Dataset Illustration

In this section, we provide a more detailed illustration of our datasets.

The CIFAR-10 dataset comprises 50,000 images across 10 classes. We designate the first 6 classes as seen classes and divide them into 3 tasks, each encompassing 2 classes. For these 6 classes, we further split data from them into labeled and unlabeled subsets. In our experiment, we adopt two different division ratios for data from seen classes: 20% labeled (thus, 80% unlabeled) and 50% labeled (equally, 50% unlabeled). For example, with a 20% labeling ratio, each class includes 1,000 labeled and 4,000 unlabeled instances, so $|D_l|$ is 6,000. We then maintain the unlabeled dataset $D_u$ using the unlabeled instances from the seen classes and all data from the 4 unknown classes, totaling 44,000 instances. Similarly, with a 50% labeling ratio, each class has 2,500 labeled and 2,500 unlabeled instances, leading to $D_l$ with 15,000 labeled instances and $D_u$ with 35,000 unlabeled instances.

For the CIFAR-100 dataset, which includes 50,000 images across 100 classes, the first 80 classes are treated as seen classes and divided into 16 tasks with five classes each. Under a 20% labeled and 80% unlabeled ratio, there are 8,000 labeled instances and 32,000 unlabeled instances across 80 seen classes. The corresponding unlabeled dataset $D_u$ consists of 32,000 unlabeled instances from 80 seen classes and 10,000 instances from 20 unknown classes.

The Tiny ImageNet contains 100,000 images of 200 classes (500 for each class). We split the first 120 classes into 20 tasks, each containing 6 classes. Under a 20% labeled and 80% unlabeled ratio, we have 12,000 labeled instances and 48,000 unlabeled instances. The unlabeled dataset $D_u$ consists of 48,000 unlabeled instances from 120 seen classes and 40,000 instances from 80 unknown classes.

During training, for each task $i$, we simultaneously sample data from the labeled dataset $\mathcal{D}_l^i$ for the current task $i$ and the shuffled unlabeled dataset $D_u$. $D_u$ consists of data from all classes, including previous task classes, current task classes, future task classes (whose labels have not been revealed and are thus treated as novel classes for the current task $i$), and unknown classes that are not included in the continual learning tasks. In each iteration, we sample both labeled and unlabeled data for each batch, adhering to the respective proportions of labeled and unlabeled data in the datasets. For example, in the CIFAR-10 dataset with a 50% labeling ratio, where we have 15,000 labeled instances and 35,000 unlabeled instances, we maintain this proportion in our sampling approach for each iteration. Consequently, in a single batch, we sample 10 labeled instances and 23 unlabeled instances. For each task, we access 5,000 labeled instances from 2 classes, and 11,500 instances from 10 classes. This approach ensures that each unlabeled sample is utilized only once in the online continual learning process.

### B.4 Baselines

In this paper, we adopt the following methods as baselines:

1. *Joint*: It gives an upper bound given by training all tasks jointly.

2. *Single* (Lopez-Paz & Ranzato, 2017): It sequentially trains a single network across all tasks.

3. *Independent* (Lopez-Paz & Ranzato, 2017): It trains multiple networks; each is trained independently for specific task.

4. *GEM* (Lopez-Paz & Ranzato, 2017): Gradient Episodic Memory (GEM) maintains an episodic memory to store samples from previous tasks and ensure the gradients for new tasks do not interfere with learned tasks.

5. *iCaRL* (Rebuffi et al., 2017): iCaRL uses a nearest-exemplar method and distillation to maintain a set of exemplars for each class.

6. *GSS* (Aljundi et al., 2019): Gradient-based sample selection(GSS) selects and replays a subset of diverse data based on the gradient to solve online continual learning.

7. *ER* (Chaudhry et al., 2019): Experience Replay (ER) trains both incoming data and data from the replay memory. Despite its simplicity, ER surpasses many advanced continual learning methods.

8. *DER* (Buzzega et al., 2020): Dark Experience Replay(DER) stores examples with their outputs, and minimizes the difference between outputs from the current model and memory.

9. *ER-ACE* (Caccia et al., 2022): ER-ACE deploys asymmetric cross-entropy for online continual learning problem.

10. *DVC* (Gu et al., 2022): DVC improves representations with contrastive learning for online continual learning. We extend their contrastive learning module to our setting.

11. *DistillMatch* (Smith et al., 2021): DistillMatch is a distillation-based method that considers SSCL by rejecting samples that are not seen in CL tasks. It uses each data more than once to train the model and OOD detector. To adapt DistillMatch to online continual learning, we provide the ground truth for OOD samples, assisting in their exclusion.

12. *ORCA* (Cao et al., 2022): ORCA is an open-World semi-Supervised learning method which recognizes previously seen classes and discovers novel classes at the same time. We combine *ORCA* with *ER* to adapt it to the CL setting.

13. *AutoNovel* (Han et al., 2020): AutoNovel is designed for the novel class discovery problem by first training on the labeled dataset and then transferring to the unlabeled dataset to discover novel classes using rank statistics. We adapt its unlabeled data learning method to our setting.

14. *FACT* (Zhou et al., 2022): FACT reserves the embedding space for new classes in future tasks to achieve forward compatibility. Considering its idea to prepare for future tasks is related to our work, we also adapt this method to our setting and make comparison.

15. *Refresh* (Wang et al., 2024): Refresh learning operates by initially unlearning current data and subsequently relearning it, which effectively enhances the learning process. We augment *ER* with refresh learning.

16. *VR-MCL* (Wu et al., 2024): VR-MCL extends meta continual learning (Gupta et al., 2020) by introducing variance reduction to achieve both timely and accurate Hessian approximation.

These methods include simple ERM methods like *Single* and *Independent* to establish basic performance baselines; continual learning (CL) methods such as *GEM*, *iCaRL*, *GSS*, *ER*, *DER*, and *Refresh* to evaluate OpenACL against regular CL approaches; state-of-the-art OCL methods like *ER-ACE* and *DVC*, which specifically address the challenges of the OCL problem; and novel class-related methods such as *DistillMatch*, *AutoNovel*, *FACT*, and *ORCA*, considering their relevance in handling novel class scenarios. For novel class discovery methods like *AutoNovel* and *FACT* that require a pre-training phase, we utilized SimCLR to pre-train the models.

### B.5 Computation and Parameter Usage

Here, we present the number of parameters used in each method in Table 7. OpenACL maintains additional proxies for unseen classes, with the parameter count for each proxy equaling the representation dimension in latent space. We also evaluate the time required for a batch update, with the results detailed in Table 8. Note that the reported time is solely for a single update iteration and does not account for memory replay.

Table 7: The number of model parameters for different datasets.

|  | OpenACL | Refresh | DVC | Others |
|---|---|---|---|---|
| CIFAR-10 | 1094740 | 2188212 | 1096544 | 1094106 |
| CIFAR-100 | 1109140 | 2212040 | 1136948 | 1106020 |
| Tiny-ImageNet | 1125140 | 2224920 | 1158788 | 1112460 |

Table 8: Average computation time for one update.

| | Refresh | DVC | DistillMatch | ORCA | OpenACL | Others |
|---|---|---|---|---|---|---|
| Time / ms | $145.9_{\pm4.36}$ / $139.2_{\pm5.26}$ / $98.9_{\pm2.63}$ | $82.5_{\pm2.50}$ | $73.8_{\pm5.13}$ | $87.6_{\pm3.75}$ | $76.4_{\pm1.96}$ | $84.6_{\pm2.01}$ / $75.5_{\pm3.49}$ / $29.1_{\pm2.66}$ |

## C   Additional Experiments

### C.1   Experiments of Multi-epoch Training

In the previous experiments, we focused on the challenging online continual learning setting to better simulate the dynamic environments where the data stream continuously evolves. However, OpenACL is capable of the general case of continual learning. Here, we also conduct experiments to compare our method with two best baselines where we train for 10 epochs on each task, instead of just 1 epoch. Each task is not revisited. Results in Tables 9 and 10 show that OpenACL consistently outperforms others when trained with multiple epochs.

Table 9: Average accuracy over three runs of multiple epochs training on Task-IL benchmarks.

| Method | CIFAR-100 | | Tiny-ImageNet | |
|---|---|---|---|---|
| Labels % | 20 | 50 | 20 | 50 |
| ER-ACE | $57.2_{\pm1.41}$ / $57.0_{\pm1.58}$ / $55.7_{\pm2.53}$ | $64.8_{\pm1.12}$ / $64.5_{\pm1.36}$ / $64.4_{\pm1.16}$ | $36.9_{\pm0.71}$ / $36.2_{\pm1.41}$ / $36.7_{\pm0.73}$ | $42.2_{\pm0.59}$ / $42.1_{\pm0.62}$ / $41.2_{\pm1.06}$ |
| DVC | $63.2_{\pm1.26}$ | $68.7_{\pm0.86}$ | $43.6_{\pm1.03}$ | $47.1_{\pm0.90}$ |
| **OpenACL** | $65.7_{\pm1.60}$ | $72.7_{\pm0.37}$ | $46.3_{\pm1.52}$ | $49.8_{\pm0.52}$ |

Table 10: Average accuracy over three runs of multiple epochs training on Class-IL benchmarks.

| Method | CIFAR-100 | | Tiny-ImageNet | |
|---|---|---|---|---|
| Labels % | 20 | 50 | 20 | 50 |
| ER-ACE | $9.6_{\pm2.28}$ / $10.2_{\pm0.90}$ / $9.7_{\pm4.30}$ | $19.3_{\pm0.70}$ / $18.9_{\pm0.47}$ / $20.4_{\pm0.53}$ | $6.3_{\pm0.37}$ / $5.2_{\pm0.24}$ / $6.8_{\pm0.61}$ | $7.4_{\pm0.29}$ / $6.1_{\pm2.25}$ / $7.6_{\pm0.23}$ |
| DVC | $18.2_{\pm1.96}$ | $24.1_{\pm1.21}$ | $8.8_{\pm0.77}$ | $11.7_{\pm1.37}$ |
| **OpenACL** | $22.9_{\pm0.86}$ | $27.0_{\pm1.02}$ | $10.2_{\pm0.44}$ | $13.6_{\pm0.84}$ |

### C.2   Experiments of Labeled/unlabeled Ratio

In this part, we study the effect of the labeled/unlabeled data ratio by varying this ratio within extreme cases to study how well the proposed method works in different scenarios. We conduct the new experiments on 10% labeled data and 80% labeled data in the Task-IL setting and present results in Table 11. Experiments with 10% labeled data evaluate the model's performance in highly constrained labeled data scenarios (e.g. 50 labeled images per class in CIFAR-100).

Table 11: Average accuracy with varying ratio.

| Method | CIFAR-100 | | Tiny-ImageNet | |
|---|---|---|---|---|
| Labels % | 10 | 80 | 10 | 80 |
| ER-ACE | $51.0_{\pm0.47}$ / $51.7_{\pm0.58}$ / $50.3_{\pm0.71}$ | $64.9_{\pm0.71}$ / $64.1_{\pm0.69}$ / $63.8_{\pm0.75}$ | $32.9_{\pm1.42}$ / $33.7_{\pm0.66}$ / $32.3_{\pm1.59}$ | $44.3_{\pm0.65}$ / $44.9_{\pm0.70}$ / $44.0_{\pm0.60}$ |
| DVC | $53.2_{\pm2.02}$ | $64.6_{\pm0.49}$ | $35.4_{\pm0.36}$ | $45.6_{\pm0.58}$ |
| **OpenACL** | $55.2_{\pm1.17}$ | $68.5_{\pm0.37}$ | $38.2_{\pm0.68}$ | $48.4_{\pm1.04}$ |

### C.3   Experiments on Stanford Cars Dataset

In this section, we further evaluate our model on Stanford Cars Dataset Krause et al. (2013) from Semantic Shift Benchmark Vaze et al. (2022b). We use the same class split as the Semantic Shift Benchmark where there are 98 known classes and 98 unknown classes. The 98 known classes are split into 14 tasks. All images are resized to $224 \times 224$ and ResNet-18 is used as the backbone. As the number of images is relatively limited, we only split the data from known classes into 50% labeled and 50% unlabeled. We train 10 epochs for each task. The results are presented in Table 12.

Table 12: Average accuracy on Stanford Cars.

| Method | Stanford Cars (Task-IL) | Stanford Cars (Class-IL) |
|---|---|---|
| ER-ACE | $17.0_{\pm 0.85}$ / $16.8_{\pm 0.28}$ / $16.7_{\pm 0.57}$ | $3.1_{\pm 0.25}$ / $3.2_{\pm 0.31}$ / $2.9_{\pm 0.21}$ |
| DVC | $17.0_{\pm 0.98}$ | $3.8_{\pm 0.25}$ |
| ORCA | $18.3_{\pm 1.16}$ | $4.3_{\pm 0.36}$ |
| **OpenACL** | $20.8_{\pm 1.02}$ | $8.0_{\pm 0.45}$ |

## C.4 Experiments on Proxy Initialization

In implementation, we use uniform initialization to initialize the proxies. To study how initialization methods affect the final performance, we here consider multiple different initialization ways, including Xavier Uniform Initialization, Kaiming Uniform Initialization, and normal initialization. From table 13, the choice of initialization does not significantly affect overall model performance.

Table 13: Average accuracy with different initialization methods in Task-IL.

| Method | CIFAR-100 | | Tiny-ImageNet | |
|---|---|---|---|---|
| Labels % | 20 | 50 | 20 | 50 |
| Uniform | $60.4_{\pm 1.19}$ | $66.6_{\pm 0.28}$ | $40.2_{\pm 0.45}$ | $47.0_{\pm 0.42}$ |
| Xavier Uniform | $60.6_{\pm 0.86}$ | $66.8_{\pm 0.66}$ | $40.0_{\pm 0.47}$ | $46.9_{\pm 0.71}$ |
| Kaiming Uniform | $60.8_{\pm 0.72}$ | $66.5_{\pm 0.40}$ | $41.1_{\pm 1.53}$ | $46.8_{\pm 0.69}$ |
| Normal | $60.8_{\pm 0.73}$ | $66.3_{\pm 0.59}$ | $40.1_{\pm 1.59}$ | $47.3_{\pm 1.39}$ |

## C.5 Experiments on Hybrid Datasets

In this experiment, we investigate model performance in a more open-world setting by incorporating unlabeled data from different datasets. This setting better reflects real-world scenarios where unlabeled data may not always show similar pattern as labeled data. Specifically, for CIFAR-100, we consider SVHN and Tiny-ImageNet as external sources of unlabeled data. Similarly, for Tiny-ImageNet, we use CIFAR-100 and SVHN as external datasets. We evaluate two configurations for incorporating external unlabeled data:

- External-only: The unlabeled data is solely from the external dataset.
- Mixed-source: The unlabeled data is a combination of the original dataset and an external dataset.

To control for dataset size and avoid bias due to sample count, we match the number of samples between datasets where necessary. For the external-only setup, if the external dataset has more samples than the original, we randomly sample the same number of examples as in the original unlabeled set. For the mixed-source setting, we augment the original unlabeled set with an additional 25% of its size using external data (e.g., CIFAR-100 + SVHN).

Table 14 shows the accuracy, backward transfer (BWT), and forward transfer (FWT) for various combinations of labeled and unlabeled data. Unlabeled data is still helpful to the continual learning tasks during the training. However, the quality and distribution of unlabeled data are critical. External datasets closer in distribution (e.g., CIFAR-100 and Tiny-ImageNet) can improve performance or maintain it. Using only mismatched data (e.g., SVHN) as unlabeled input can lead to noticeable degradation compared with training on original unlabeled datasets, however, it is still better than a purely supervised learning scenario as shown Table in 4.

## C.6 Proxy-Class Assignment Precision Analysis

We analyzed proxy-to-class assignment precision during continual learning. Specifically, we calculated the precision as the proportion of samples mapped to a given proxy that belongs to the class to those samples that the proxy is ultimately assigned. This metric reflects how well a proxy captures class-specific representation during training. Table 15 reports the average proxy precision ($\pm$ standard deviation) across tasks for both Task-IL and Class-IL settings on CIFAR-100 and Tiny-ImageNet. We also report the lower bound, which

Table 14: Average performance with different unlabeled datasets in Task-IL.

| | CIFAR-100 | | | Tiny-ImageNet | | | |
| --- | --- | --- | --- | --- | --- | --- | --- |
| **Unlabeled** | Acc | BWT | FWT | **Unlabeled** | Acc | BWT | FWT |
| CIFAR-100 | $66.6_{\pm0.28}$ | $9.2_{\pm1.65}$ | $13.0_{\pm1.48}$ | Tiny-ImageNet | $47.0_{\pm0.42}$ | $2.7_{\pm1.36}$ | $10.9_{\pm1.10}$ |
| CIFAR-100 + SVHN | $65.9_{\pm1.16}$ | $9.0_{\pm2.06}$ | $12.3_{\pm1.95}$ | Tiny + SVHN | $47.7_{\pm0.92}$ | $2.7_{\pm0.44}$ | $10.1_{\pm0.32}$ |
| CIFAR-100 + Tiny | $66.3_{\pm0.79}$ | $8.8_{\pm1.45}$ | $10.7_{\pm2.42}$ | Tiny + CIFAR-100 | $47.5_{\pm0.35}$ | $2.5_{\pm0.46}$ | $9.0_{\pm0.72}$ |
| SVHN | $63.7_{\pm0.84}$ | $7.3_{\pm2.03}$ | $8.7_{\pm0.59}$ | SVHN | $44.3_{\pm0.73}$ | $1.4_{\pm1.26}$ | $5.6_{\pm1.28}$ |
| Tiny | $66.7_{\pm1.06}$ | $8.4_{\pm0.92}$ | $11.2_{\pm1.22}$ | CIFAR-100 | $45.4_{\pm0.23}$ | $1.6_{\pm0.79}$ | $8.6_{\pm1.79}$ |

corresponds to the expected precision under uniform random assignment (20% for CIFAR-100 (5 classes/task) and 16.7% for Tiny-ImageNet (6 classes/task).

Table 15: Precision of the proxy adaptation.

| | **CIFAR-100** | **Tiny-ImageNet** |
| --- | --- | --- |
| **Task-IL** | $61.4_{\pm3.67}$ | $51.7_{\pm1.27}$ |
| **Class-IL** | $57.8_{\pm2.23}$ | $47.2_{\pm0.51}$ |
| **Lower Bound** | 20 | 16.7 |

# D   Supplementary Results

Here, we present the full version of Table 1 and 2 in Table 18 and 19. We also reported Averaged Anytime Accuracy (average accuracy after each task) in Table 20 and 21. Note that some baselines such as AutoNovel and FACT require a pretraining phase, which allows these methods to learn meaningful patterns early in the training process, potentially giving them an unfair advantage in AAA evaluation compared to methods trained from scratch.

## D.1   Ablation studies in Class-IL

We extend our ablation studies to the Class-IL setting and report results in Table 16 and Table 17. Note that, we do not report Forward Transfer (FWT) in the Class-IL setting. This is because, before observing new tasks during training, the model's predictions are inherently biased toward the output channels corresponding to previously seen tasks. As a result, FWT in Class-IL does not provide a reliable reflection of the model's ability to generalize to future tasks.

Table 16: Ablation study on the proxy adaptation. We report average accuracy over three runs using different variants of OpenACL in Class-IL.

| | **CIFAR-100** | | **Tiny-ImageNet** | |
| --- | --- | --- | --- | --- |
| | **Acc** | **BWT** | **Acc** | **BWT** |
| **w/o PA** | $19.3_{\pm1.97}$ | $-21.0_{\pm1.50}$ | $11.8_{\pm1.20}$ | $-18.1_{\pm1.36}$ |
| **w/o K** | $19.8_{\pm1.67}$ | $-21.6_{\pm1.79}$ | $12.1_{\pm0.85}$ | $-16.1_{\pm0.75}$ |
| **w/o A** | $18.9_{\pm0.56}$ | $-21.2_{\pm1.04}$ | $11.1_{\pm0.50}$ | $-17.8_{\pm1.65}$ |
| **OpenACL** | $20.0_{\pm1.23}$ | $-21.0_{\pm2.25}$ | $11.9_{\pm1.06}$ | $-17.6_{\pm0.75}$ |

Table 17: Ablation study on the number of proxies in Class-IL.

| | CIFAR-100 | | | Tiny-ImageNet | |
|---|---|---|---|---|---|
| Proxies | 20 | 50 | Proxies | 20 | 50 |
| 90 | $15.1_{\pm1.01}$ | $19.4_{\pm0.83}$ | 150 | $8.8_{\pm0.42}$ | $13.2_{\pm0.86}$ |
| 100 | $15.7_{\pm0.44}$ | $20.0_{\pm1.23}$ | 200 | $7.9_{\pm0.37}$ | $11.9_{\pm1.06}$ |
| 200 | $14.2_{\pm0.97}$ | $18.0_{\pm0.91}$ | 300 | $7.9_{\pm1.36}$ | $11.9_{\pm0.13}$ |
| 300 | $13.2_{\pm1.57}$ | $16.5_{\pm0.52}$ | 400 | $7.5_{\pm0.22}$ | $11.7_{\pm1.03}$ |
| Incremental | $15.6_{\pm1.62}$ | $20.2_{\pm1.48}$ | Incremental | $8.1_{\pm1.26}$ | $13.0_{\pm0.38}$ |

Table 18: Table 1 with standard deviation.

| Method | CIFAR-10 | | CIFAR-100 | | Tiny-ImageNet | |
|---|---|---|---|---|---|---|
| Labels % | 20 | 50 | 20 | 50 | 20 | 50 |
| Joint | $68.3_{\pm0.60}$ / $68.9_{\pm0.93}$ / $67.9_{\pm0.80}$ | $69.1_{\pm1.22}$ / $69.4_{\pm1.25}$ / $68.7_{\pm1.94}$ | $68.4_{\pm0.26}$ / $68.1_{\pm0.53}$ / $67.5_{\pm0.94}$ | $76.6_{\pm1.27}$ / $75.7_{\pm0.55}$ / $75.1_{\pm0.79}$ | $52.8_{\pm1.01}$ / $50.3_{\pm0.35}$ / $50.7_{\pm0.33}$ | $58.3_{\pm0.97}$ / $57.8_{\pm0.66}$ / $57.0_{\pm1.76}$ |
| Single | $57.5_{\pm3.67}$ / $57.6_{\pm3.49}$ / $54.7_{\pm2.54}$ | $59.3_{\pm2.78}$ / $57.0_{\pm1.83}$ / $57.6_{\pm2.05}$ | $33.5_{\pm1.27}$ / $34.1_{\pm3.10}$ / $32.3_{\pm2.48}$ | $37.9_{\pm2.82}$ / $36.3_{\pm2.63}$ / $37.2_{\pm1.61}$ | $20.9_{\pm1.99}$ / $20.5_{\pm0.69}$ / $19.6_{\pm0.54}$ | $25.9_{\pm1.14}$ / $23.3_{\pm0.83}$ / $23.1_{\pm1.16}$ |
| Independent | $62.5_{\pm3.22}$ / $64.2_{\pm1.35}$ / $61.3_{\pm2.56}$ | $63.9_{\pm3.69}$ / $62.3_{\pm2.43}$ / $62.5_{\pm2.83}$ | $26.7_{\pm3.98}$ / $30.3_{\pm3.28}$ / $31.8_{\pm2.88}$ | $36.2_{\pm2.30}$ / $36.2_{\pm2.15}$ / $33.4_{\pm1.67}$ | $21.6_{\pm0.83}$ / $21.5_{\pm1.07}$ / $23.2_{\pm1.72}$ | $26.5_{\pm0.84}$ / $28.0_{\pm2.21}$ / $27.0_{\pm1.79}$ |
| iCaRL | $56.0_{\pm1.07}$ / $57.4_{\pm1.38}$ / $56.7_{\pm2.19}$ | $57.2_{\pm1.35}$ / $58.7_{\pm0.97}$ / $58.3_{\pm2.20}$ | $45.8_{\pm1.50}$ / $45.9_{\pm2.68}$ / $46.4_{\pm0.58}$ | $44.1_{\pm1.38}$ / $42.3_{\pm1.70}$ / $41.8_{\pm1.09}$ | $25.2_{\pm1.03}$ / $25.3_{\pm1.75}$ / $23.5_{\pm1.39}$ | $31.3_{\pm1.01}$ / $29.0_{\pm1.72}$ / $26.5_{\pm2.71}$ |
| DER | $62.2_{\pm0.71}$ / $63.9_{\pm3.30}$ / $63.3_{\pm2.09}$ | $63.2_{\pm2.58}$ / $63.9_{\pm2.42}$ / $63.6_{\pm2.39}$ | $38.6_{\pm3.03}$ / $38.7_{\pm2.51}$ / $39.6_{\pm3.24}$ | $46.8_{\pm1.92}$ / $44.7_{\pm2.36}$ / $44.0_{\pm2.82}$ | $24.2_{\pm2.64}$ / $22.4_{\pm2.68}$ / $25.8_{\pm1.02}$ | $28.4_{\pm2.24}$ / $29.6_{\pm2.27}$ / $28.0_{\pm1.66}$ |
| GEM | $61.3_{\pm1.08}$ / $64.0_{\pm2.24}$ / $62.6_{\pm2.18}$ | $63.2_{\pm0.82}$ / $63.6_{\pm2.39}$ / $64.2_{\pm0.52}$ | $53.5_{\pm1.38}$ / $52.6_{\pm0.79}$ / $51.8_{\pm0.82}$ | $58.6_{\pm1.57}$ / $57.5_{\pm1.59}$ / $54.4_{\pm1.67}$ | $33.0_{\pm1.07}$ / $35.4_{\pm1.56}$ / $32.1_{\pm1.49}$ | $40.1_{\pm2.10}$ / $37.3_{\pm1.20}$ / $38.0_{\pm2.35}$ |
| ER | $62.9_{\pm1.17}$ / $62.3_{\pm3.32}$ / $61.3_{\pm3.58}$ | $64.9_{\pm3.88}$ / $63.8_{\pm6.12}$ / $63.7_{\pm1.09}$ | $54.8_{\pm1.74}$ / $55.3_{\pm0.65}$ / $53.7_{\pm1.09}$ | $59.9_{\pm2.87}$ / $58.5_{\pm1.39}$ / $57.8_{\pm0.84}$ | $35.2_{\pm0.55}$ / $36.3_{\pm1.79}$ / $35.2_{\pm0.15}$ | $41.7_{\pm0.34}$ / $41.4_{\pm0.39}$ / $40.2_{\pm0.10}$ |
| ER-ACE | $61.2_{\pm1.83}$ / $61.6_{\pm3.78}$ / $61.3_{\pm2.45}$ | $62.4_{\pm0.91}$ / $64.2_{\pm2.95}$ / $63.9_{\pm1.99}$ | $53.8_{\pm2.08}$ / $55.0_{\pm0.78}$ / $54.8_{\pm1.78}$ | $61.7_{\pm0.71}$ / $62.4_{\pm0.93}$ / $62.1_{\pm0.86}$ | $36.2_{\pm1.36}$ / $37.2_{\pm0.78}$ / $35.4_{\pm1.25}$ | $41.4_{\pm0.54}$ / $42.4_{\pm1.63}$ / $40.6_{\pm0.74}$ |
| Refresh | $63.0_{\pm2.25}$ / $63.1_{\pm3.04}$ / $61.7_{\pm1.31}$ | $62.6_{\pm1.91}$ / $64.3_{\pm2.23}$ / $62.6_{\pm2.70}$ | $54.7_{\pm2.71}$ / $55.3_{\pm0.52}$ / $55.1_{\pm0.20}$ | $61.2_{\pm1.18}$ / $61.9_{\pm1.26}$ / $61.0_{\pm1.17}$ | $35.8_{\pm0.87}$ / $36.9_{\pm1.16}$ / $35.8_{\pm0.38}$ | $42.6_{\pm0.23}$ / $42.1_{\pm1.51}$ / $41.5_{\pm0.55}$ |
| VR-MCL | $60.3_{\pm1.82}$ | $63.4_{\pm3.42}$ | $53.3_{\pm1.13}$ | $63.3_{\pm0.84}$ | $32.9_{\pm0.25}$ | $40.3_{\pm1.70}$ |
| DVC | $57.4_{\pm0.86}$ | $61.7_{\pm3.23}$ | $57.6_{\pm0.92}$ | $62.7_{\pm2.08}$ | $36.8_{\pm0.61}$ | $43.5_{\pm0.35}$ |
| DistillMatch | $57.8_{\pm6.45}$ | $59.4_{\pm1.67}$ | $35.7_{\pm1.78}$ | $41.3_{\pm1.96}$ | $21.8_{\pm0.49}$ | $26.2_{\pm2.05}$ |
| AutoNovel | $56.3_{\pm1.82}$ | $56.5_{\pm2.11}$ | $58.7_{\pm0.13}$ | $63.3_{\pm0.83}$ | $37.4_{\pm0.74}$ | $43.1_{\pm4.74}$ |
| FACT | $53.2_{\pm3.27}$ | $55.3_{\pm1.78}$ | $55.9_{\pm2.86}$ | $62.8_{\pm1.00}$ | $35.0_{\pm1.49}$ | $42.3_{\pm0.67}$ |
| ORCA | $60.9_{\pm1.93}$ | $62.2_{\pm2.13}$ | $56.4_{\pm1.17}$ | $62.4_{\pm0.68}$ | $34.4_{\pm1.19}$ | $39.3_{\pm0.95}$ |
| **OpenACL** | $\mathbf{64.3}_{\pm2.75}$ | $\mathbf{66.3}_{\pm1.17}$ | $\mathbf{60.4}_{\pm1.19}$ | $\mathbf{66.6}_{\pm0.28}$ | $\mathbf{40.2}_{\pm0.45}$ | $\mathbf{47.0}_{\pm0.42}$ |

Table 19: Table 2 with standard deviation.

| Method | CIFAR-100 | | Tiny-ImageNet | |
|---|---|---|---|---|
| Labels % | 20 | 50 | 20 | 50 |
| Joint | $22.8_{\pm0.80}$ / $23.0_{\pm0.64}$ / $21.8_{\pm0.48}$ | $31.8_{\pm2.09}$ / $32.9_{\pm0.85}$ / $30.8_{\pm1.60}$ | $13.4_{\pm0.83}$ / $14.4_{\pm0.31}$ / $13.6_{\pm1.28}$ | $22.0_{\pm2.25}$ / $21.5_{\pm1.40}$ / $21.1_{\pm0.88}$ |
| Single | $3.1_{\pm0.20}$ / $2.8_{\pm0.20}$ / $2.5_{\pm0.09}$ | $3.0_{\pm0.37}$ / $2.5_{\pm0.69}$ / $3.0_{\pm0.31}$ | $1.9_{\pm0.09}$ / $2.0_{\pm0.11}$ / $1.7_{\pm0.12}$ | $2.4_{\pm0.12}$ / $2.8_{\pm0.27}$ / $2.7_{\pm0.13}$ |
| iCaRL | $6.8_{\pm1.19}$ / $7.0_{\pm0.56}$ / $6.3_{\pm1.25}$ | $7.3_{\pm0.66}$ / $8.3_{\pm0.50}$ / $7.0_{\pm0.96}$ | $4.5_{\pm0.95}$ / $3.3_{\pm0.19}$ / $3.4_{\pm0.30}$ | $4.1_{\pm0.29}$ / $4.8_{\pm0.36}$ / $4.2_{\pm0.31}$ |
| DER | $3.7_{\pm0.11}$ / $3.7_{\pm0.23}$ / $3.5_{\pm0.31}$ | $3.6_{\pm0.23}$ / $3.9_{\pm0.57}$ / $3.9_{\pm0.81}$ | $2.4_{\pm0.11}$ / $2.5_{\pm0.13}$ / $2.1_{\pm0.19}$ | $2.4_{\pm0.10}$ / $2.6_{\pm0.16}$ / $2.3_{\pm0.27}$ |
| GEM | $7.0_{\pm0.14}$ / $8.0_{\pm0.47}$ / $6.9_{\pm1.48}$ | $9.7_{\pm1.06}$ / $7.7_{\pm2.15}$ / $6.7_{\pm2.27}$ | $2.4_{\pm0.08}$ / $3.4_{\pm0.24}$ / $2.7_{\pm0.17}$ | $2.3_{\pm0.66}$ / $2.6_{\pm0.09}$ / $1.8_{\pm0.44}$ |
| GSS | $12.8_{\pm0.64}$ / $11.2_{\pm0.32}$ / $10.3_{\pm1.28}$ | $16.8_{\pm1.11}$ / $15.3_{\pm2.27}$ / $15.2_{\pm1.54}$ | $3.3_{\pm0.21}$ / $5.4_{\pm0.63}$ / $3.8_{\pm0.33}$ | $5.3_{\pm0.40}$ / $5.6_{\pm0.36}$ / $5.0_{\pm0.13}$ |
| ER | $10.9_{\pm0.71}$ / $12.0_{\pm0.84}$ / $11.5_{\pm1.38}$ | $15.6_{\pm0.93}$ / $15.8_{\pm0.98}$ / $16.9_{\pm0.45}$ | $3.3_{\pm0.06}$ / $4.2_{\pm0.46}$ / $3.9_{\pm0.15}$ | $4.8_{\pm0.22}$ / $6.7_{\pm0.61}$ / $5.7_{\pm0.31}$ |
| ER-ACE | $12.8_{\pm0.20}$ / $13.3_{\pm0.90}$ / $12.0_{\pm0.79}$ | $16.7_{\pm0.79}$ / $17.9_{\pm0.63}$ / $17.1_{\pm1.20}$ | $5.0_{\pm0.55}$ / $5.4_{\pm0.56}$ / $4.9_{\pm0.36}$ | $7.4_{\pm0.74}$ / $8.1_{\pm0.90}$ / $7.2_{\pm0.52}$ |
| Refresh | $10.6_{\pm0.57}$ / $11.6_{\pm1.58}$ / $11.2_{\pm0.92}$ | $16.9_{\pm0.20}$ / $18.1_{\pm1.00}$ / $17.3_{\pm1.20}$ | $5.2_{\pm0.48}$ / $5.5_{\pm0.22}$ / $5.4_{\pm0.74}$ | $6.6_{\pm0.46}$ / $7.3_{\pm0.39}$ / $6.9_{\pm0.30}$ |
| VR-MCL | $12.3_{\pm1.14}$ | $17.4_{\pm0.53}$ | $5.2_{\pm0.17}$ | $7.5_{\pm0.80}$ |
| DVC | $11.2_{\pm0.78}$ | $16.2_{\pm2.08}$ | $5.8_{\pm0.40}$ | $8.3_{\pm1.42}$ |
| DistillMatch | $2.8_{\pm0.06}$ | $3.2_{\pm0.17}$ | $2.0_{\pm0.18}$ | $2.7_{\pm0.14}$ |
| AutoNovel | $13.2_{\pm0.61}$ | $17.9_{\pm1.19}$ | $6.5_{\pm0.57}$ | $9.2_{\pm0.58}$ |
| FACT | $12.9_{\pm0.84}$ | $16.3_{\pm0.89}$ | $5.9_{\pm0.90}$ | $8.2_{\pm1.18}$ |
| ORCA | $14.4_{\pm0.37}$ | $18.8_{\pm0.52}$ | $6.8_{\pm0.57}$ | $9.6_{\pm1.20}$ |
| **OpenACL** | $\mathbf{15.7}_{\pm0.44}$ | $\mathbf{20.0}_{\pm1.23}$ | $\mathbf{7.9}_{\pm0.37}$ | $\mathbf{11.9}_{\pm1.06}$ |

Table 20: Averaged Anytime Accuracy (AAA) over three runs of experiments on Task-IL benchmarks.

| Method | CIFAR-10 | | CIFAR-100 | | Tiny-ImageNet | |
|---|---|---|---|---|---|---|
| Labels % | 20 | 50 | 20 | 50 | 20 | 50 |
| Single | $63.1_{\pm2.23}$ / $63.3_{\pm1.76}$ / $60.6_{\pm3.72}$ | $61.1_{\pm4.79}$ / $62.0_{\pm0.95}$ / $62.6_{\pm1.68}$ | $29.8_{\pm1.88}$ / $36.1_{\pm1.87}$ / $29.4_{\pm1.63}$ | $32.7_{\pm3.98}$ / $31.0_{\pm1.80}$ / $39.3_{\pm2.63}$ | $19.9_{\pm1.66}$ / $20.0_{\pm0.48}$ / $19.2_{\pm1.34}$ | $24.9_{\pm2.14}$ / $20.8_{\pm0.22}$ / $22.2_{\pm1.48}$ |
| Independent | $65.9_{\pm3.88}$ / $66.9_{\pm2.53}$ / $63.9_{\pm3.22}$ | $64.6_{\pm4.74}$ / $59.4_{\pm2.99}$ / $60.8_{\pm3.25}$ | $25.5_{\pm3.40}$ / $28.9_{\pm2.18}$ / $29.8_{\pm1.51}$ | $32.8_{\pm1.76}$ / $33.8_{\pm3.94}$ / $31.1_{\pm1.73}$ | $20.8_{\pm1.09}$ / $21.0_{\pm1.28}$ / $22.4_{\pm1.63}$ | $24.5_{\pm1.57}$ / $25.2_{\pm2.94}$ / $24.9_{\pm2.07}$ |
| iCaRL | $63.9_{\pm2.26}$ / $61.0_{\pm5.44}$ / $60.9_{\pm3.13}$ | $59.6_{\pm3.25}$ / $57.6_{\pm5.43}$ / $56.7_{\pm3.50}$ | $43.7_{\pm0.89}$ / $44.0_{\pm1.15}$ / $43.6_{\pm0.52}$ | $40.9_{\pm1.22}$ / $40.6_{\pm1.39}$ / $40.6_{\pm0.90}$ | $25.1_{\pm1.53}$ / $21.2_{\pm1.19}$ / $21.8_{\pm1.74}$ | $27.5_{\pm1.09}$ / $28.3_{\pm1.06}$ / $24.5_{\pm2.28}$ |
| DER | $65.0_{\pm2.44}$ / $65.8_{\pm1.80}$ / $65.4_{\pm2.79}$ | $63.7_{\pm2.99}$ / $64.0_{\pm3.33}$ / $63.6_{\pm3.44}$ | $37.9_{\pm1.88}$ / $38.2_{\pm2.12}$ / $37.0_{\pm0.92}$ | $45.8_{\pm1.36}$ / $44.6_{\pm1.10}$ / $44.9_{\pm1.30}$ | $23.4_{\pm2.46}$ / $21.7_{\pm1.12}$ / $23.8_{\pm1.44}$ | $27.2_{\pm1.08}$ / $26.6_{\pm1.18}$ / $27.4_{\pm1.40}$ |
| GEM | $65.1_{\pm0.76}$ / $65.9_{\pm1.97}$ / $64.0_{\pm1.97}$ | $62.9_{\pm2.97}$ / $62.6_{\pm3.76}$ / $63.5_{\pm1.27}$ | $44.7_{\pm0.96}$ / $42.5_{\pm0.91}$ / $41.6_{\pm1.01}$ | $50.9_{\pm1.21}$ / $50.7_{\pm1.59}$ / $51.3_{\pm1.09}$ | $27.6_{\pm0.91}$ / $27.4_{\pm2.96}$ / $27.3_{\pm0.79}$ | $34.9_{\pm0.82}$ / $33.1_{\pm0.91}$ / $33.8_{\pm0.05}$ |
| ER | $62.9_{\pm1.63}$ / $63.9_{\pm2.27}$ / $63.7_{\pm1.63}$ | $61.2_{\pm0.70}$ / $60.3_{\pm0.32}$ / $60.3_{\pm0.42}$ | $45.2_{\pm0.12}$ / $46.5_{\pm0.17}$ / $47.2_{\pm0.41}$ | $53.3_{\pm0.40}$ / $52.8_{\pm0.98}$ / $55.6_{\pm1.37}$ | $30.1_{\pm0.51}$ / $29.3_{\pm1.99}$ / $30.5_{\pm0.25}$ | $35.4_{\pm0.77}$ / $35.3_{\pm0.63}$ / $35.4_{\pm0.68}$ |
| ER-ACE | $62.2_{\pm1.74}$ / $63.5_{\pm2.31}$ / $65.7_{\pm1.00}$ | $60.5_{\pm3.50}$ / $62.9_{\pm2.27}$ / $60.8_{\pm1.38}$ | $46.5_{\pm0.81}$ / $45.9_{\pm1.43}$ / $46.1_{\pm1.74}$ | $52.7_{\pm0.58}$ / $52.7_{\pm0.97}$ / $53.0_{\pm0.56}$ | $30.4_{\pm0.60}$ / $30.7_{\pm0.35}$ / $29.5_{\pm2.49}$ | $35.5_{\pm0.44}$ / $35.9_{\pm0.46}$ / $33.8_{\pm2.29}$ |
| Refresh | $65.6_{\pm1.96}$ / $65.2_{\pm2.44}$ / $63.7_{\pm3.49}$ | $61.2_{\pm3.41}$ / $61.9_{\pm0.45}$ / $59.1_{\pm1.25}$ | $46.2_{\pm0.97}$ / $45.2_{\pm1.43}$ / $45.4_{\pm1.11}$ | $51.7_{\pm0.84}$ / $52.1_{\pm0.96}$ / $52.1_{\pm0.36}$ | $30.1_{\pm0.86}$ / $30.8_{\pm0.24}$ / $29.8_{\pm0.88}$ | $34.6_{\pm2.16}$ / $34.1_{\pm1.30}$ / $35.4_{\pm1.26}$ |
| VR-MCL | $62.6_{\pm2.52}$ | $59.0_{\pm5.66}$ | $43.1_{\pm0.33}$ | $53.0_{\pm1.47}$ | $28.3_{\pm0.42}$ | $35.8_{\pm1.70}$ |
| DVC | $61.6_{\pm3.57}$ | $61.3_{\pm2.91}$ | $49.3_{\pm0.97}$ | $54.3_{\pm0.78}$ | $30.6_{\pm0.28}$ | $37.4_{\pm0.73}$ |
| DistillMatch | $62.0_{\pm2.68}$ | $59.6_{\pm3.72}$ | $29.2_{\pm5.03}$ | $38.1_{\pm3.21}$ | $19.3_{\pm0.40}$ | $24.7_{\pm1.12}$ |
| AutoNovel | $61.5_{\pm3.75}$ | $62.4_{\pm1.07}$ | $43.2_{\pm2.03}$ | $58.7_{\pm0.89}$ | $32.8_{\pm3.02}$ | $37.3_{\pm3.58}$ |
| FACT | $54.4_{\pm3.13}$ | $51.6_{\pm6.59}$ | $\mathbf{54.7}_{\pm0.53}$ | $\mathbf{59.4}_{\pm0.94}$ | $28.9_{\pm1.40}$ | $39.1_{\pm0.36}$ |
| ORCA | $62.7_{\pm4.54}$ | $60.1_{\pm2.90}$ | $47.2_{\pm1.41}$ | $55.7_{\pm0.55}$ | $30.9_{\pm1.26}$ | $34.8_{\pm0.63}$ |
| **OpenACL** | $\mathbf{67.6}_{\pm0.57}$ | $\mathbf{65.8}_{\pm3.33}$ | $52.8_{\pm1.32}$ | $58.9_{\pm0.79}$ | $\mathbf{34.9}_{\pm0.88}$ | $\mathbf{39.9}_{\pm0.82}$ |

Table 21: Averaged Anytime Accuracy (AAA) over three runs of experiments on Class-IL benchmarks.

| Method | CIFAR-100 | | Tiny-ImageNet | |
|---|---|---|---|---|
| Labels % | 20 | 50 | 20 | 50 |
| Single | $7.9_{\pm0.39}$ / $7.9_{\pm0.54}$ / $7.0_{\pm0.58}$ | $9.1_{\pm0.66}$ / $8.7_{\pm0.81}$ / $8.9_{\pm0.46}$ | $4.4_{\pm0.55}$ / $4.7_{\pm0.43}$ / $4.4_{\pm0.64}$ | $5.6_{\pm0.87}$ / $5.8_{\pm0.33}$ / $5.5_{\pm0.18}$ |
| iCaRL | $13.9_{\pm0.82}$ / $13.5_{\pm0.79}$ / $12.4_{\pm0.82}$ | $15.8_{\pm0.57}$ / $15.5_{\pm0.28}$ / $14.9_{\pm0.55}$ | $7.9_{\pm0.27}$ / $7.5_{\pm0.57}$ / $7.4_{\pm0.32}$ | $9.1_{\pm0.26}$ / $9.4_{\pm0.26}$ / $9.1_{\pm0.13}$ |
| DER | $8.8_{\pm0.81}$ / $8.5_{\pm0.59}$ / $8.2_{\pm0.15}$ | $11.7_{\pm0.16}$ / $12.2_{\pm1.63}$ / $12.7_{\pm1.11}$ | $4.4_{\pm0.47}$ / $4.8_{\pm0.33}$ / $4.0_{\pm0.64}$ | $6.3_{\pm0.22}$ / $6.7_{\pm0.38}$ / $6.1_{\pm0.45}$ |
| GEM | $15.0_{\pm1.93}$ / $15.3_{\pm2.09}$ / $11.9_{\pm2.42}$ | $14.8_{\pm2.94}$ / $13.2_{\pm1.08}$ / $12.9_{\pm1.59}$ | $6.3_{\pm0.41}$ / $7.5_{\pm0.98}$ / $7.3_{\pm0.69}$ | $7.3_{\pm0.47}$ / $7.6_{\pm0.26}$ / $6.7_{\pm0.37}$ |
| GSS | $17.0_{\pm1.19}$ / $16.6_{\pm1.03}$ / $16.2_{\pm1.46}$ | $26.0_{\pm0.50}$ / $25.3_{\pm0.76}$ / $25.6_{\pm0.26}$ | $9.1_{\pm0.31}$ / $10.8_{\pm0.68}$ / $9.3_{\pm0.57}$ | $13.1_{\pm0.49}$ / $13.3_{\pm0.25}$ / $13.4_{\pm0.20}$ |
| ER | $20.0_{\pm1.10}$ / $20.3_{\pm0.22}$ / $20.1_{\pm1.43}$ | $25.0_{\pm2.23}$ / $25.6_{\pm0.48}$ / $26.3_{\pm0.87}$ | $8.7_{\pm0.30}$ / $10.5_{\pm0.08}$ / $8.8_{\pm0.56}$ | $12.0_{\pm0.26}$ / $14.3_{\pm0.07}$ / $13.1_{\pm0.16}$ |
| ER-ACE | $21.4_{\pm0.41}$ / $21.2_{\pm0.53}$ / $19.4_{\pm0.47}$ | $25.1_{\pm0.89}$ / $26.2_{\pm0.67}$ / $25.4_{\pm0.86}$ | $9.7_{\pm0.30}$ / $11.1_{\pm0.25}$ / $9.0_{\pm1.14}$ | $13.5_{\pm0.43}$ / $13.9_{\pm0.91}$ / $12.6_{\pm0.49}$ |
| Refresh | $19.5_{\pm0.20}$ / $20.3_{\pm0.36}$ / $19.7_{\pm0.16}$ | $26.0_{\pm0.38}$ / $26.8_{\pm0.77}$ / $26.4_{\pm0.74}$ | $10.3_{\pm0.18}$ / $11.0_{\pm0.64}$ / $9.6_{\pm0.83}$ | $13.6_{\pm0.53}$ / $15.0_{\pm0.65}$ / $14.0_{\pm0.64}$ |
| VR-MCL | $15.0_{\pm1.19}$ | $24.0_{\pm1.31}$ | $8.0_{\pm0.96}$ | $12.6_{\pm0.14}$ |
| DVC | $17.7_{\pm1.25}$ | $21.9_{\pm0.20}$ | $9.2_{\pm0.13}$ | $12.2_{\pm0.57}$ |
| DistillMatch | $8.8_{\pm0.48}$ | $10.6_{\pm1.36}$ | $4.6_{\pm0.34}$ | $5.5_{\pm0.20}$ |
| AutoNovel | $22.7_{\pm0.30}$ | $\mathbf{28.9}_{\pm0.75}$ | $12.0_{\pm0.91}$ | $15.9_{\pm0.24}$ |
| FACT | $\mathbf{24.8}_{\pm0.25}$ | $28.6_{\pm0.79}$ | $12.8_{\pm0.39}$ | $16.3_{\pm0.20}$ |
| ORCA | $23.9_{\pm0.26}$ | $27.6_{\pm0.50}$ | $12.5_{\pm0.42}$ | $16.1_{\pm0.88}$ |
| **OpenACL** | $22.8_{\pm1.06}$ | $26.7_{\pm2.19}$ | $\mathbf{12.9}_{\pm0.88}$ | $\mathbf{16.5}_{\pm0.36}$ |

