# OpenReview forum: "Exploiting Open-World Data for Adaptive Continual Learning"
_TMLR — Rejected by TMLR_

### Review · Reviewer_CApu · 2025-02-12

**Summary Of Contributions:**

This paper investigates continual learning in open-world environments, where the learner has access to a limited number of labeled samples from known classes and a large pool of unlabeled data that may contain unknown classes. The main contribution is to develop a method that can leverage unlabeled data to enhance learning performance. Unlike previous approaches that typically filter out unknown classes from the unlabeled data, this work proposes utilizing the entire unlabeled dataset to learn class proxies, including those for unknown classes. The authors show that incorporating the knowledge of unknown classes can improve future predictions, particularly when future known classes are present within the unlabeled data as the unknown classes for the current task. Extensive experiments validate the effectiveness of the proposed method.

**Audience:**

Yes

**Claims And Evidence:**

No

**Requested Changes:**

- Could you provide real-world examples where the unlabeled dataset contains the full set of classes?
- Could you offer a more detailed discussion on how the proposed method helps minimize Eq. (1)?
- The proposed method appears to require sampling from the entire unlabeled dataset $\mathcal{D}_u$. Given that $\mathcal{D}_u$ is assumed to be large in this setting, I am concerned about the storage cost of the proposed method compared to other approaches.
- The definitions of the open space $\mathcal{O}$ and $\mathcal{S}$ are not entirely clear. It is unclear how $\mathcal{S}$ and $\mathcal{O}$ are formally defined based on empirical samples.
- It is also unclear how the initialization of the proxies affects the method’s performance. The paper would be clearer if the authors provided details on how the proxies are initialized and discussed how the initialization impacts empirical performance.
- In the paragraph "Evaluation on unlabeled data," the authors demonstrate that the unlabeled dataset remains beneficial even when the unknown classes in it are irrelevant to the tasks. Could you provide a more detailed explanation for this observation and discuss whether unlabeled data is always helpful? I wonder if this effect arises because the experimental environment is relatively static, with datasets across tasks sharing the same style. It would be interesting to evaluate the proposed method in a hybrid setting—just for instance, where some tasks come from MNIST and others from SVHN.

**Strengths And Weaknesses:**

The strengths of the paper are as follows:
- The open-world continual learning problem is interesting, and the idea of leveraging unknown classes to improve future predictions is novel to me.
- The paper is well-structured and clearly written for the most part.
- Extensive experiments are conducted to validate the proposed method, including comparisons with various existing approaches in the literature and ablation studies to demonstrate the effectiveness of each component.

The weaknesses of the paper are as follows:
- **On the performance measure:** While the paper is well-organized overall, the exact performance measure of the proposed method remains unclear to me. From an algorithmic perspective, it appears that the objective is to minimize Eq. (1). However, the paper provides only an intuitive discussion at the end of Section 4.2, which may not be sufficient for clarity. Additionally, Eq. (1) seems to be defined over the dataset $\mathcal{D}_{\ell}^i$ rather than the underlying distribution. This raises concerns about whether minimizing it is a reasonable objective, given the limited availability of labeled data, and whether the chosen performance measure could lead to overfitting. From an empirical standpoint, it is also unclear how accuracy is defined—whether it considers only known classes or also accounts for the ability to identify unknown classes.

- **On the problem setup:** While the big picture of open-world continual learning is convincing, the paper assumes that the unlabeled dataset contains the full set of classes and is available at every iteration of the learning process. This seems like a strong assumption, as it allows the learner to access information about future tasks, potentially making the continual learning problem less challenging. I think it would be necessary to provide a concrete example illustrating why this problem setting is practical.

---

> ### Author Response · Authors · 2025-03-25
>
> Dear Reviewer CApu,
>
> We thank the reviewer for the evaluation of our work. We address the reviewer's concerns below:
>
> ## Real-world examples where the unlabeled dataset contains the full set of classes? (also weakness 2)
>
> An example is that **User-generated content platforms**: On platforms like YouTube, Instagram, or e-commerce websites, new content is constantly uploaded by users. Even though labels may be missing or incomplete, the data itself often includes instances of emerging or trending categories before they are formally introduced into the system. For example, new tech products may appear in user-uploaded content long before they are manually labeled and added as new categories. In addition, for seen classes,  we can still train from user-uploaded data or internet data but we only have a limited number of samples to perform supervised training due to the cost the human labeling. In such cases, the unlabeled dataset naturally includes a mixture of samples from past tasks (seen classes), future tasks (unseen classes), and inevitably some unknown classes.
>
> We encountered this exact scenario in an object detection application development, where we received limited labeled data from a data provider, but the majority of the dataset was unlabeled. Due to limited labeling resources, only a subset of predefined classes was annotated, while user-uploaded content may not align with these predefined categories. The unlabeled data—though only roughly selected—contained a combination of samples from predefined classes and some novel classes as user-uploaded that required model adaptation.
>
> ## Could you offer a more detailed discussion on how the proposed method helps minimize Eq. (1)?
>
> We have updated our method section (Section 4) to give more discussion. Specifically, proxy contrastive learning aligns labeled data with their proxies to compact the seen region $\mathcal{S}_t$  and encourages novel or out-of-distribution data to be mapped distinctly, instead of being falsely assigned to high-confidence regions associated with seen classes. Therefore, it reduces the probability mass assigned to novel samples in regions of seen classes, thereby minimizing the open risk.
>
> ## The storage cost of the proposed method compared to other approaches
>
> We store the unlabeled dataset on disk rather than loading the entire dataset into memory. During training, we dynamically sample only the required data points from the disk as needed. It reduces memory usage and allows scalability to large datasets without incurring significant system overhead.
>
> In terms of cost, the storage requirement is negligible relative to the hardware investment typically needed for deep learning. For example, high-end GPUs such as the NVIDIA A100 (80GB) or H100 range from $10,000 to $30,000 per unit, and large-scale training often involves multiple such GPUs. In contrast, high-capacity storage devices—such as a 16TB HDD ($300–$400) or even enterprise-grade SSDs—are significantly more affordable.
>
> ## The definitions of the open space $\mathcal{O}$ and $\mathcal{S}$ and the performance measure in weakness 1.
>
> We have updated our problem formulation section (Section 3) to clarify our objectives. As we are using unlabeled data to improve the continual learning on labeled datasets, we report the accuracy for the continual learning tasks.
>
> ##  It is also unclear how the initialization of the proxies affects the method’s performance.
>
> At the beginning of training, we use uniform initialization with PyTorch's spectral normalization to initialize the proxies. However, this initialization primarily impacts only the first task, as subsequent tasks utilize Proxy Adaptation, which updates the proxy weights based on their nearest samples. This adaptation ensures that proxies continuously represent the evolving data distribution.
>
> To address potential concerns about initialization methods, we have conducted additional experiments comparing various initialization strategies, including Xavier uniform initialization, Kaiming uniform initialization, and normal initialization. The experimental results (included in Appendix C.4) indicate that the choice of initialization does not significantly affect overall model performance.
>
> ### Evaluate the proposed method in a hybrid setting
>
> To address your concern, we consider new experiments in Appendix C.5 where the unlabeled dataset consists of data from a different dataset, or is entirely consisting of samples from a different dataset. For instance, in the CIFAR-100 experiments, we considered two scenarios: (i) the unlabeled data is a mix of CIFAR-100 and SVHN, and (ii) the unlabeled data is solely from SVHN.  We found that while performance tends to degrade when the unlabeled data distribution is significantly different, the model still benefits from the additional data compared to the supervised-only baseline.

---

### Review · Reviewer_GoPi · 2025-02-22

**Summary Of Contributions:**

The paper proposes a new setting for continuous learning where the model is also given a set of unlabeled samples that do not align with any of the continuous learning tasks. The paper proposes a method that learns the proxies for both the labeled and unlabeled points using contrastive learning. When new tasks come during continual learning, the proposed method utilizes the most relevant proxies and the corresponding samples to get more information for the semi-supervised learning procedure. The paper shows that it gets improved performance in such a new setting compared to baselines.

**Audience:**

Yes

**Claims And Evidence:**

Yes

**Requested Changes:**

Please address the weaknesses.

**Strengths And Weaknesses:**

Strengths:
1. The paper proposes a new setting for continual learning that is closer to the real-world scenario.

2. The paper proposes an algorithm that can effectively utilize the information in the unknown samples to aid continual learning tasks. The paper shows improved performance compared to baseline methods.

3. The number of proxies can be dynamically adjusted, which makes the method more widely applicable.


Weaknesses:
1. The experimental setting is not that ''open'' compared to the claims made in the paper, as the unknown class samples are obtained by dividing classes of relatively small vision datasets. Moreover, the results only show one division for one dataset (e.g., 6 known and 4 unknown for CIFAR10), which I think is insufficient to demonstrate the effectiveness of the proposed method and investigate behaviors of various methods in the proposed new setting. Ideally, a more extensive dataset should be used to mimic the ``openness'' of the real-world setting.

2. The proposed method introduces several new hyper-parameters. The total loss combines three losses, each with a temperature parameter. Even though the paper uses certain standard values for the temperature hyper-parameters, and there are no weights for combining the three losses, it is unclear whether these values require adjustment for other datasets as the datasets shown in the experiments are all well-explored.

3. The new setting does not entirely convince me. If we don't care about the classes, i.e., the unknown samples, those samples are typically available in large quantities. Therefore, we could apply a different approach and pretrain models to learn good representations before the task-specific continual learning. Particularly, as the paper mostly demonstrates on image classification, unlabeled images are relatively easy to obtain, and a more reasonable approach would be to utilize some pretrained image representation model, which already has some nice clustering structures. I wonder if it is possible to give a more concrete application where the unknown samples cannot be obtained so easily and we have similar known v.s. unknown divisions presented in the paper's experiments.

---

> ### Author Response · Authors · 2025-03-25
>
> Dear Reviewer GoPi,
>
> We thank the reviewer for the evaluation of our work. We address the reviewer's concerns below:
>
> ## Weaknesses 1: A more extensive dataset should be used to mimic the ``openness'' of the real-world setting.
>
> To mimic the more diverse openness of datasets, we use the different split ratios for CIFAR-100 and Tiny-Imagenet200 which makes it more challenging to train on Tiny-Imagenet since it has fewer known classes (60\% known classes ) compared with CIFAR-100 (80\% known classes). To further address this concern, we conducted additional experiments where the unlabeled dataset consists of data from a different dataset, or is entirely composed of samples from a different distribution. For example, in the CIFAR-100 experiments, we evaluated two setups: (i) an unlabeled set that includes both CIFAR-100 and SVHN samples, and (ii) an unlabeled set containing only SVHN data. These configurations simulate the heterogeneity of real-world unlabeled data sources. The results of these experiments are included in the Appendix C.5. We found that although using only mismatched data (e.g., SVHN) as the unlabeled set leads to some performance degradation compared to using in-distribution unlabeled data, it still outperforms the supervised-only baseline.  Furthermore, augmenting the original unlabeled set with a small portion of external data maintains stable performance.
>
> ## Weaknesses 2 hyper-parameters tuning
>
> We followed the default values for the temperature hyperparameters as recommended in prior work. For the loss combination, we did not perform an exhaustive study on the impact of varying the loss weights; instead, we set the weights of different loss terms to 1 across all datasets. While this simplifies the training procedure, we acknowledge that a more thorough hyperparameter search on the weighting of loss components could lead to further performance gains.
>
> ## Weaknesses 3: the new setting
>
> Pretrained models are expected to achieve good performance on unlabeled datasets, however, how to make the representation function robust against catastrophic forgetting is also challenging. For example, our method still outperforms two baselines AutoNovel and FACT, which require a pre-training phase and are pretrained on the whole dataset using SimCLR.
>
> For a more concrete application where the unknown samples cannot be obtained so easily and we have similar known v.s. unknown divisions. Medical imaging (e.g. skin disease diagnosis) is one of the examples. In this domain, collecting large-scale images is challenging due to privacy concerns, and annotation costs are exceptionally high due to the necessity of expert experience.  In this scenario, the model is initially trained on a limited set of labeled diseases, some disease categories may be missing from the training data. As new tasks arise and additional data is labeled, previously unknown diseases can be discovered and incorporated into the learning process. This setting aligns well with our study, as it reflects a real-world continual learning challenge where unknown categories emerge dynamically, and exhaustive pretraining on a vast, unlabeled dataset is extremely time-consuming. Another relevant case is training a model on user-uploaded data (e.g. new image labels, photo styles, and product categories), where unknown classes naturally exist. In such scenarios, the model is initially trained on a set of known categories, but as users continuously upload new data, novel and previously unseen classes emerge dynamically. These data are hard to be comprehensively labeled, but they are beneficial to the model training.

---

### Review · Reviewer_9TaC · 2025-03-11

**Summary Of Contributions:**

This paper proposes leveraging unlabeled data to enhance the continual learning (CL) capability of models. The authors consider the less-explored open semi-supervised continual learning (Open SSCL) setting, where unlabeled data may contain samples from current, future, or unknown classes—an important and highly applicable scenario in real-world applications. To tackle this challenge, the paper introduces OpenACL, a method that maintains learnable proxies for seen tasks and reserves additional proxies for future tasks. These reserved proxies are dynamically assigned to new classes when they are introduced in subsequent labeled tasks, enhancing the forward transfer of the method.

**Audience:**

Yes

**Claims And Evidence:**

Yes

**Requested Changes:**

- Some of the equations and variables were not too clear. What is the role of softmax temperature (s) and temperature (k) ? Some variables are used without defining/explaining them. The transition from Eqn 4 to 5 was not clear.
     - Include analysis on how well proxy creation and assignment align with true class labels - How often do samples from the same class contribute to the same proxy? How well do assigned proxies match future task labels?
     - How fair comparison is ensures for CL methods with supervised setting and non-replay methods? Also include more recent Online-CL methods and provide AAA performance as well. Add stronger baselines for non-online CL based methods as well.
     - Provide ablation studies in Class-IL, not just Task-IL.
     - Clarify the role of episodic memory in more detail and explicitly mention memory buffer sizes for baselines
     - More explanation of the methodology (expand figure1 caption in appendix?)

**Strengths And Weaknesses:**

- **Strengths**
     - The paper highlights an underexplored setting in CL, where large amounts of unlabeled data can be leveraged to learn useful representations for both current and future tasks.
     - The paper is well-organized and well-written. The motivation and the research questions are explained clearly.
     - The authors validate their method through results on multiple datasets, along with ablation studies on key components and parameters.


- **Weaknesses**
    - The setting assumes that all the classes in the new task would be present in the unlabeled dataset available in the previous dataset. This may not necessarily hold in many real world applications. So it still caters to a specific use-case and is not very generic?Thoughts?
     - Analysis
          - The study does not provide direct empirical evidence that the method correctly creates reserve proxies and assigns them to the correct class. There are intuitions and claims that will help with more analysis to understand better.
          - The intuition or claims of the proxy-level contrastive learning mechanism were not too clear. Ex. evidence for the claim of reducing intra-class variance?
     - Comparisons
          - While the authors claim to focus on Online Continual Learning (Online-CL), many of the chosen baselines are not designed for online CL. Several recent SOTA Online-CL methods are missing from the comparison, such as: [1] VR-MCL, [2] La-MAML
          - Online-CL typically reports “Average Any-Time Accuracy (AAA),” which is missing.
     - Memory -  The algorithm mentions an episodic memory, but this is not discussed in the method section. Additionally, the buffer size for the baselines is not provided in the main paper, which makes it difficult to assess comparisons
     - The ablation study (Table 5) shows that omitting proxy allocation (w/o A) significantly reduces forgetting, but the paper does not provide a reason for this behavior.

---

> ### Author Response · Authors · 2025-03-25
>
> Dear Reviewer 9TaC,
>
> We thank the reviewer for the evaluation of our work. We address the reviewer's concerns below:
>
> ## Equations and variables (Change 1)
>
> Equation 5 defines the proxy contrastive learning loss function applied to the combined labeled and unlabeled data. Specifically, Equation 4 represents the loss computed on the unlabeled data portion, which is incorporated into Equation 5 as the unlabeled loss term. We have also revised our method section to introduce the role of softmax temperature.
>
> ## The setting assumption (Weakness 1)
>
> We would like to clarify that our setting is motivated by practical deployment scenarios, particularly in online platforms where user-uploaded data often does not fit into our predefined classes (e.g. a bamboo image is classified as a "leg" because the model has never been exposed to the concept of "bamboo"). In such cases, the system must adapt with limited human supervision as it is not feasible to fully label all potential classes in advance. However, these emerging classes are not entirely inaccessible—roughly selected or weakly curated unlabeled data containing samples from such novel classes can often be obtained from data providers, the internet, or the user side. Therefore, we consider a semi-supervised continual learning paradigm where the model incrementally adapts by leveraging unlabeled data which may contain future classes, before transitioning to supervised learning when sufficient labels become available in a new task.
>
> While our setting does assume that some of the future task classes are present in earlier unlabeled data, this reflects a trade-off between fully supervised and fully open-world learning. In fact, our experiment where unlabeled data are all from novel classes and hybrid dataset experiments (Appendix C.5) where unlabeled data are from a different dataset further show that the model remains effective even when the unlabeled data doesn't follow our assumption. Thus, while not fully generic, the setting captures a realistic and valuable deployment use case that motivates the proposed approach.
>
>
> ## Analysis of how well proxy creation and assignment align with true class labels and intuition  (Weakness 2 and Changes 2)
>
> To directly evaluate whether the method correctly creates reserve proxies and assigns them to the intended classes, we conducted a quantitative analysis of the precision of proxy-to-class assignments after each task. Specifically, we measured the proportion of samples assigned to a proxy that truly belonged to all the samples it was finally assigned to, showing how accurately a prototype (proxy) represents a single task class. Our method achieves high proxy precision across both Task-IL and Class-IL settings on CIFAR-100 and Tiny-ImageNet. The results are provided in Appendix C.6. We have also updated the discussion of the proxy-level contrastive learning mechanism in section 4.2
>
>
> ## Comparison (Weakness 2 and change 3,4)
>
> Thank you for pointing out the relevant work. We found that [1] VR-MCL builds upon [2] La-MAML and is a more recent method. Therefore, we have included [1] VR-MCL as an additional baseline in our experiments and cited [2] La-MAML in the manuscript to provide context for this baseline. For non-online continual learning methods, our benchmark already includes a strong set of baselines, such as the state-of-the-art method Refresh (published in 2024). With a total of 16 baselines covered, we believe the comparison is comprehensive and fair. Additionally, we have included the results for AAA and Class-IL ablation studies in Appendix D. We would like to clarify that AAA may not be a very suitable metric in our comparison, as some baselines such as AutoNovel and FACT require a pretraining phase. It allows these methods to learn meaningful patterns early in the training process, potentially giving them an unfair advantage in AAA evaluation compared to methods trained from scratch.
>
> We have included results for ablation studies in Class-IL in Appendix D.1
>
>
> ## Memory(Weakness 4 and change 5)
>
> Regarding the buffer size, we make the buffer size consistent for all baselines to make fair comparisons. We have updated our manuscript to state that in Appendix B.1 and clarify the role of episodic memory in section 4.1.
>
>
> ## Weakness 5 and Changes 6
>
> We have added more discussion in the experiment section to address this concern. Removing proxy allocation (w/o A) may lead to incorrect associations between representation clusters from k-means and existing class proxies, making it challenging to remap representations correctly and can result in noisy updates to existing class proxies. Therefore, we can see a reduction in forgetting mitigation ability (decreasing in BWT). In addition, we have also expanded the caption in Figure 1.

---

### Decision · Action_Editor_HWGw · 2025-06-13

**Recommendation:** Reject

**Additional Comments:**

I appreciate the authors effort into putting together this work, and that the method shows improvement over existing CL work.
However, reviewers have raised concerns about the validity and practicality of the proposed setting. While the examples provided by the authors are interesting, all experiments are done on toy datasets. How can we know that the proposed method can in fact work on the mentioned examples or that the design of the setting match these hypothetical examples?
Further I don't think a performance below 20% is convincing for a practical scenario. The method does not provide theoretical insights to overlook the small scale of experiments.
I encourage the authors to add an experiment mimicking the examples mentioned e.g., medical imaging or object detection. If relevance is shown indeed, I believe this work is worth publishing. However, at this current shape of toy experiments and unrealistic setting I don't think the paper can appear at TMLR.

**Audience:**

Yes

**Audience Explanation:**

Researchers working on CL might be interested in this setting, however given the concerns about the practicality this AE is not sure that such direction is worth research efforts.

**Claims And Evidence:**

No

**Claims Explanation:**

The authors claims about the practicality of the setting and the examples given by the authors are not reflected in any experiment rather small  scale experiments on toy datasets.

**Resubmission Of Major Revision:**

The authors may consider submitting a major revision at a later time.